# Bridging Expressivity and Scalability
# with Adaptive Unitary SSMs

**Arjun Karuvally**
Salk Institute for Biological Studies
akaruvally@salk.edu

**Franz Nowak**
ETH Zürich
franz.nowak@inf.ethz.ch

**T. Anderson Keller**
The Kempner Institute for the Study of Natural
and Artificial Intelligence at Harvard University
t.anderson.keller@gmail.com

**Carmen Amo Alonso**
Computer Science Department
Stanford University
camoalon@stanford.edu

**Terrence J. Sejnowski**
Salk Institute for Biological Studies
terry@salk.edu

**Hava T. Siegelmann**
University of Massachusetts Amherst
hava@umass.edu

## Abstract

Recent work has revealed that state space models (SSMs), while efficient for long-sequence processing, are fundamentally limited in their ability to represent formal languages—particularly due to time-invariant and real-valued recurrence structures. In this work, we draw inspiration from adaptive and structured dynamics observed in biological neural systems and introduce the Adaptive Unitary State Space Model (AUSSM): a novel class of SSMs that leverages skew-symmetric, input-dependent recurrence to achieve unitary evolution and high expressive power. Using algebraic automata theory, we prove that AUSSM can perform modulo counting and simulate solvable group automata at precision logarithmically bounded in the input length, enabling SSMs to model a broad class of regular languages out of reach for other SSM architectures. To overcome the practical inefficiencies of adaptive recurrence, we develop a separable convolution formulation and a CUDA implementation that enables scalable parallel training. Empirically, we show that AUSSM and its hybrid variant—interleaved with Mamba—outperform prior SSMs on formal algorithmic tasks such as parity and modular arithmetic, and achieve competent performance on real-world long time-series classification benchmarks. Our results demonstrate that adaptive unitary recurrence provides a powerful and efficient inductive bias for both symbolic and continuous sequence modeling. The code is available at https://github.com/arjunkaruvally/AUSSM

## 1 Introduction

Modeling long-range dependencies efficiently and expressively remains a central challenge in sequence modeling. While Transformer architectures have achieved state-of-the-art results across domains such as language modeling [1, 2, 3], forecasting [4, 5], and protein design [6], their quadratic complexity with respect to sequence length limits scalability [7]. In response, recent work has explored *state space models* [8, 9] (SSMs) as a scalable alternative, using linear-time convolutions and structured recurrence to enable efficient processing of long sequences [9, 10]. Despite the computational advantages SSMs offer, they are fundamentally limited in their ability to express general linear time-varying systems and formal languages efficiently. Even basic regular languages

that require counting, such as parity or balanced parentheses [11] are impossible for practically used SSM architectures like Mamba that have positive real eigenvalue spectra. Frontier SSMs are either Linear Time Invariant (LTI) or partially Linear Time Varying (LTV) [1], resulting in weaker expressivity compared to more general LTV systems that are capable of approximating any non-linear dynamical systems [12]. Two properties thus emerge as necessary for increasing the expressivity of SSMs: a general eigenvalue spectrum and adaptive recurrence. However, incorporating both these properties naively in SSMs introduces gradient instability through the exploding/vanishing gradient problem [13], and leads to quadratic computational complexity, severely limiting scalability.

In this work, we propose the *Adaptive Unitary State Space Model* (AUSSM) as a principled middle ground between scalability and expressivity. AUSSM is a **fully adaptive state space model** with linear time-varying recurrence and a unitary eigenvalue spectrum, combining the theoretical benefits of time-varying recurrence with the practical advantages of structured, conserved dynamics. We formally prove that AUSSM can perform modulo counting with constant-width hidden states, and that combining AUSSM with existing non-adaptive models like Mamba yields *maximal expressivity within the class of diagonal SSMs*. To make this architecture scalable, *we introduce a novel separable kernel formulation for adaptive SSMs* that exposes efficient computational algorithms which reduce the quadratic cost of adaptive recurrence to linear time and space. Empirically, we validate the theoretical claims through a suite of algorithmic tasks, demonstrating substantial performance gains over Mamba, and showing that AUSSM retains competitive efficiency through an optimized CUDA implementation. Further, we evaluate the long-range modeling capabilities by testing on a suite of time series benchmarks.

Interestingly, structured unitary and adaptive dynamics are also found as emergent behavior in biological neural systems [14] and even trained non-linear recurrent neural networks [15], where they are believed to support flexible integration of information over space and time [16]. We take the computational role of these structured unitary dynamics as a motivation to derive AUSSM using a skew-symmetric ODE used in identifying purely rotational features from data in neuroscience [17].

AUSSM provides a new architectural foundation that bridges formal expressivity and practical scalability (Figure 1). It expands the space of scalable SSMs by showing that adaptive (and time-varying) recurrences can be made computationally efficient, unlocking new capabilities for symbolic and long-context sequence modeling that is grounded in biological principles and theory.

## 2 Background and Motivation

State Space Models (SSMs) have emerged as efficient alternatives to Transformers for sequence modeling, particularly in long-context settings. SSMs compress arbitrarily long sequences into a fixed-dimensional hidden state using a recurrent formulation and this can be computed in parallel using an efficient convolution formulation.

The most general SSMs are described by a continuous-time Ordinary Differential Equation (ODE):

$$\frac{dx(t)}{dt} = A_t x(t) + B_t u(t), \quad y(t) = C_t x(t) \tag{1}$$

or its discrete counterpart:

$$x(t) = A'_t x(t-1) + B'_t u(t), \quad y(t) = C'_t x(t) \tag{2}$$

where $x(t) \in \mathbb{R}^n$ is the hidden state, $u(t)$ is the input, and $y(t)$ is the output. The matrices $A_t, B_t, C_t$ define the system dynamics and may vary over time. In the discrete system, these matrices have an equivalent discretized counterpart in $A', B', C'$, respectively. The discrete recurrence can be reformulated as a parallel convolution:

$$y(t) = \sum_{k \leq t} C'_t \left( A'_{t-1} \cdots A'_{k+1} \right) B'_k u(k) \tag{3}$$

However, this convolution kernel requires $\mathcal{O}(L^2)$ memory for sequence length $L$, as each kernel entry must be stored for $t$ and $k$ that index over the sequence length. To avoid this, most practical SSMs

---

[1]Interested readers are referred to Appendix B, Expressivity of Single Block SSMs for the definitions of LTI, partial LTV, and LTV.

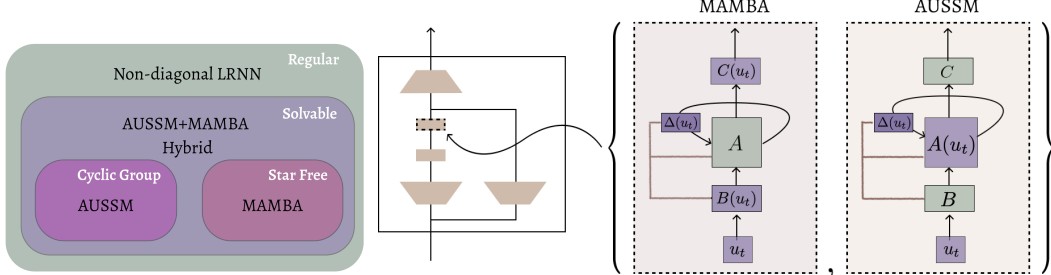

(a) Formal language classes recogniz-
able by different architectures.

(b) SSM block structure of Mamba and AUSSM (input-dependent com-
ponents are shaded in blue).

Figure 1: **(a)** Existing practical SSM blocks like Mamba use fast parallel algorithms for computing the output, resulting in a tradeoff with expressivity. Non-diagonalizable Linear RNNs are the most expressive (in formal language terms) but lack scalable computational algorithms and suffer from gradient issues. AUSSM balances the expressivity-scalability tradeoff using a fully adaptive diagonal unitary recurrence. Fast SSMs with improved expressivity can be built by combining AUSSM with MAMBA blocks. **(b)** The AUSSM block uses the same block structure as Mamba [10], where the S6 SSM in Mamba is replaced with AUSSM. The main difference between AUSSM and S6 is the adaptive recurrence, where in the case of S6, $B$, $C$, and $\Delta$ are adaptive, whereas in AUSSM, $\Delta$ and $A$ are adaptive (see Section 3 for details). AUSSM blocks can be used as drop-in replacements for existing SSM backbones to provide higher expressivity (see Section 3.1 for theoretical and Section 5 for experimental validation).

assume time-invariant dynamics ($A_k = A, B_k = B, C_k = C$), allowing for a compressed storage of the kernel but significantly restricting expressivity. Recent SSMs like Mamba [10] introduce *partial adaptivity*, where $B, C$, and step size $\Delta$) are adaptive while keeping $A$ fixed or diagonal. However, such models cannot simulate general Linear Time-Varying (LTV) systems or perform counting-based tasks (e.g., parity, modular arithmetic) with constant-width hidden states (see Appendix B). These limitations prevent Mamba from modeling input-sensitive dynamics or general multiscale time-varying behavior (Appendix B). There are other approaches that try to improve the expressivity of SSMs by generalizing real-valued recurrences. Notably, Linear Recurrent Units [18] generalize the real-valued eigenvalue spectra with initialization close to the unit circle on the imaginary plane. This formulation has been shown to be capable of universal approximation when interleaved with non-linear multi-layer perceptrons [19]. However, this approximation relies on perfectly storing the dynamical system history without regard to resource constraints. General LTV systems are much more flexible as they have the capability to gate information based on input, thereby retaining only selected information (compressed information) that is necessary for processing, rather than a lossless history. Another notable work is linear Oscillatory State Spaces (linOSS) [20], where simple harmonic ODEs are discretized to derive novel oscillatory SSMs with conservation properties identical to AUSSM. The linOSS models are more expressive than SSMs with purely real eigenvalues, but fall short of an LTV system. AUSSM balances the two - the improved expressivity of diagonal LTV systems (using adaptive recurrence) and the scalability of separable convolutions.

## 3   Adaptive Unitary State Space Model (AUSSM)

We tackle the problem of balancing expressivity with scalability in Adaptive State Space Models by introducing two features. Adaptive input-dependent recurrent matrix improving expressivity, and unitary dynamics addressing training scalability. In this section, we derive AUSSM from the skew-symmetric ODE used to identify rotational features in the brain [17], then we prove that the inputs control AUSSM rotational frequencies smoothly, enabling a stable and effective adaptive SSM. Next, we prove that the AUSSM, combined with regular Mamba layers, is maximally expressive in the class of diagonal SSMs in terms of formal language recognition.

To derive the AUSSM model with purely rotational properties, we use the skew-symmetric Ordinary Differential Equation (ODE) used in the rotational Principal Component Analysis (jPCA) procedure - a variant of Principal Component Analysis (PCA) used in neuroscience [17]. jPCA is used to identify

rotational features of a dynamical system using observations from it. Since our requirement is to process an input signal $u(t)$ into the hidden state, we use a version of the jPCA ODE with control input given by Equation 1, with the additional constraint that the input matrix $A_t$ is skew-symmetric (with purely imaginary eigenvalues) and $B_t$ and $C_t$ stay constant with time, i.e., $B_t = B$ and $C_t = C$. We discretize the ODE following the procedure used in Mamba [10] with a step size $\Delta_t \in \mathbb{R}$ to obtain a discrete dynamical system (See Appendix C for details),

$$\begin{cases} x(t) = \exp(\Delta_t A_t)\, x(t-1) \, + \Delta_t B\, u(t)\,, \\ y(t) = C\, x(t)\,. \end{cases} \tag{4}$$

Note here that $A_t$ changes with time from adaptivity, and it is a skew-symmetric matrix. We assume that $A_t, \forall t$ belongs to a class of simultaneously diagonalizable matrices. Therefore $\exp(\Delta_t A_t)$ can be diagonalized to obtain $\exp(\Delta_t i\Lambda_j(t))$ where $\Lambda_j(t) \in \mathbb{R}$ and each $i\Lambda_j(t)$ is the $j^{\text{th}}$ eigenvalue of the matrix $A_t$. This implies that the final discrete dynamical system has purely unitary eigenvalues, i.e., eigenvalues exactly on the unit circle. The AUSSM ODE is a marginally stable, time-varying linear system where the input both drives and dynamically reshapes the system. The skew-symmetric nature of $A_t$ guarantees marginal stability by ensuring that all eigenvalues lie on the imaginary axis in continuous time, or on the unit circle after discretization (see Lem. 4 in Appendix D). This structure enables long-term memory retention without gradient explosion or decay (see Lem. 5 in Appendix D)[2].

The adaptivity of $A_t$ is enforced by making $A_t$ a function of input with $A_t = f(u(t))$ where $f : \mathbb{R} \to \mathbb{R}^n$ is the function defining how the input influences the recurrent matrix. With adaptivity, the input acts as a control signal, shaping the rotational dynamics based on the instantaneous input, analogous to gain scheduling or bilinear control systems [21, 22]. This design allows the system to dynamically traverse a spectrum of rotational behaviors in the state space, facilitating expressive temporal modeling driven by the input signal.

**Theorem 1** (Input-Modulated Rotation Frequencies via Skew-Symmetric Generator). *Let $A : \mathbb{R} \to \mathbb{R}^{n \times n}$ be a smooth function such that $A(u)$ is skew-symmetric for all $u \in \mathbb{R}$. Then for each $u \in \mathbb{R}$, all eigenvalues of $A(u)$ lie on the imaginary axis, and the eigenvalues of the discrete-time transition matrix $\Phi(u) = \exp(\Delta A(u))$ lie on the complex unit circle. Furthermore, the eigenvalues of $A(u)$ depend continuously on $u$, and thus the angular frequency of state-space rotation is smoothly and directly modulated by the input. See proof in Appendix D.*

Hence, by designing $A(u)$ appropriately (e.g., via a learnable function $f(u)$), the AUSSM can modulate the rotational speed and mode structure of the hidden state space based on the input signal in a smooth and controlled manner. Further details on how the above SSM is practically implemented are in Section 4, where we diagonalize the above ODE and introduce input adaptivity. The inputs also have a dimension of $d$, in which case the proposed SSM is applied to each input dimension following the approach used in Mamba.

### 3.1 Formal Expressivity

*Given our construction of the AUSSM, how expressive is it for formal languages?*

The goal of a formal expressivity theory is to determine, for a given architecture class, which functions or formal languages can be represented by some finite instantiation of that architecture. The quantification is over architectures —i.e., over possible finite hyperparameter settings such as model dimension, input dimension, or transition rank—rather than over the parameters within a fixed instantiation. Formal expressivity analysis goes beyond our earlier discussion of limitations of SSMs in expressing LTV dynamical systems (further detailed in Appendix B).

Representing simple formal languages has been found to be a major weakness of SSMs. Recently, a flurry of research has utilized algebraic automata theory, specifically Krohn-Rhodes theory [23], to analyze the types of formal languages expressible by different LLM architectures, notably transformers [24] and SSMs [11, 25]. The Krohn–Rhodes decomposition theorem states that any finite-state machine can be simulated by a cascade of simpler automata drawn from two types: permutation automata, which model reversible group-like behavior, and reset automata, which model state-resetting dynamics (the next state depends only on the input, not on the current state). This result implies

---

[2]In practice, there will always be small deviations from the ideal theoretical behavior due to the limited precision of modern computers.

that complex regular languages can be recognized by composing SSMs that simulate these simple automata. There is a subset of finite-state automata whose decomposition contains only set-reset automata and *cyclic* permutation automata, which suffices to recognize a large subset of regular languages, the so-called solvable languages.

Most SSMs used in practice have diagonal or diagonalizable transition matrices $A$ that can only have positive eigenvalues. As [11] showed, this means they cannot perform modulo counting, restricting their expressivity to the subset of star-free regular languages.[3] [11] also outlines the necessary conditions for SSMs to overcome this limitation and represent a larger class of regular languages. This requires 1. the ability to implement modulo counters, and 2. the ability to implement Krohn-Rhodes cascade products. Here, we reiterate their relevant results and show that our implementation not only satisfies these conditions but can recognize any solvable regular language, a language class out of reach for most practical SSMs. For a unified overview situating our expressivity results within the SSM expressivity literature, see §A.

**Fact 1** ([11], Thm. 2). *Diagonal (or diagonalizable) SSMs with only positive eigenvalues cannot perform modulo counting at finite precision, which means they can only recognize star-free languages.[4]*

**Lemma 1.** *For any $k \in \mathbb{Z}^+$, one can construct a single-layer AUSSM that counts modulo $k$, which means AUSSMs can simulate arbitrary cyclic group automata.*

*Proof sketch.* Assume we want to count the number of 1's modulo 2 in a length-T input sequence $(u)_{t=1,\ldots,T} \in \{0,1\}^T$. A single-layer AUSSM with $x_0 = 1$, $A(1) = -1$, $A(0) = 1$, and $B(0) = B(1) = 0$ will have a hidden state of $x_t = -1$ for odd counts and $x_t = 1$ for even counts of 1 up to position $t$. Similarly, to count modulo 4, we can set $A(1)$ to the fourth root of unity, i.e., either $i$ or $-i$, and $A(0) = 1, B(0) = B(1) = 0$, as before. This method can be extended to other mod $k$ counters by setting $A(1)$ to the $k$th root of unity, $\exp(2\pi i/k)$. An AUSSM can take on these parameters as it uses input-dependent A matrices whose eigenvalues lie on the unit circle of the complex plane.[5] This technique can be extended to perform modulo $k$ addition, which allows the simulation of cyclic group automata (see §E). □

**Lemma 2.** *An SSM consisting of interleaved Mamba and AUSSM blocks (hybrid Mamba+AUSSM) can implement cascade products of automata simulated by Mamba SSMs and AUSSMs.*

*Proof sketch.* [11, Lem. 19] showed that multilayer Mamba SSMs can implement cascade products of Mamba layers simulating set-reset automata, which, by Schützenberger's theorem [26], means they can recognize any star-free language. This can easily be extended to show that any automaton simulated by Mamba or AUSSM layers can be joined into a cascade product within alternating Mamba and AUSSM blocks. This works because we can always add additional padding layers at any point in the hybrid SSM without changing the behavior of the remainder of the SSM. □

**Theorem 2.** *Hybrid Mamba+AUSSM can recognize any solvable language, that is, any regular language whose syntactic monoid does not contain non-solvable subgroups.*

*Proof sketch.* By Lem. 1, an AUSSM layer can simulate cyclic group automata, and [11, Lem. 19] showed that a Mamba layer can simulate set-reset automata. Now, the Krohn-Rhodes theorem states that every finite automaton divides a cascade of alternating aperiodic monoids (set-reset automata) and finite simple groups (permutation automata). A finite group is solvable iff its decomposition series contains only cyclic groups of prime order (cyclic group automata with prime-length cycles) [27, Ex. 3.4.8]. By Lem. 2, hybrid Mamba+AUSSM can implement the Krohn-Rhodes cascade product of set-reset automata (Mamba) and cyclic group automata (AUSSM). Therefore, it can recognize all solvable languages. (cf. [11, Thm. 21]). □

---

[3]Star-free languages are those languages that can be defined without the use of a Kleene star, only using concatenation, union, and complement.

[4]Note that this theorem assumes finite precision; similar limitations are expected to persist under logarithmically bounded precision, though we do not attempt a full extension here.

[5]We assume an idealized setting in which the numerical precision is logarithmic in the sequence length. For most sequence lengths seen in practice, this is a reasonable assumption (see §E for details).

Regular languages that require representing more complex non-solvable group transformations, such as the word problem in $S_5$ or $A_5$, lie outside of this set, and according to widely held assumptions about computational expressivity theory, cannot be modeled by diagonal SSMs [28]. This means combining Mamba with AUSSM maximizes the representational capacity of diagonal SSMs (short of lifting the diagonal transition restraint, which leads to poor scaling).

## 4 Separable Convolution Kernels for Scalable Adaptive SSMs

One of the main challenges in designing SSMs is the computational efficiency of the implementation. Simulating the discrete dynamical system in Equation 4 naively is not computationally efficient as it leads to quadratic memory scaling when it is parallelized using the typical SSM convolution procedure (Appendix F.1). In this section, we introduce a separable kernel formulation for the efficient computation of adaptive time-varying SSMs. Our formulation works directly in the convolution form of the SSM and instantly exposes the separability and is applicable to a wider class of adaptive SSMs as shown below. We note here that the separable kernel formulation is not specialized for the AUSSM, but **applies to any class of SSMs that are simultaneously diagonalizable** [29]. We therefore formulate the theory in the general case and provide sufficient conditions to apply the theory in practice.

The general convolution formulation of general SSMs in Equation 3 is typically used to convert a discrete dynamical system form of an SSM to an efficient parallel implementation. This form is abstracted as applying a convolution on the input, following the equation

$$y(t) = \sum_{k \leq t} K(t, k) u(k) . \tag{5}$$

The reason for the quadratic memory scaling of the convolution operation can be observed in this abstracted form as storing the $K(t, k)$ convolution kernel requires, in general, $O(L^2)$ memory, where $L$ is the sequence length.

Separable convolution kernels have the additional property that $K(t, k) = f(t) g(k)$ that enables writing the output as

$$y(t) = f(t) \sum_{k \leq t} g(k) u(k) . \tag{6}$$

Storing the additional $f(t)$ and $g(k)$ requires only an additional $O(2L)$ memory. This is comparable to the non-adaptive case, which has a scaling $O(L)$, producing asymptotic memory efficiency matching that of the non-adaptive SSM with only a constant factor increase in memory use. The above convolution formulation can be efficiently computed in $O(\log(L))$ time by using the parallel prefix sum algorithm [30]. It is instructive to apply this formulation to an existing SSM to identify efficient computational structures - we use the partially adaptive Selective State Space Model (S6) used in the popular Mamba model [10].

**Separable Convolution Formulation of Mamba Selective SSM (S6):** In S6, the matrices $C$ and $B$ vary with input (making the SSM selective to input), in addition to the step size $\Delta$ varying with time. This generalization results in the output of the SSM written in the convolution form as:

$$y_{ti} = \sum_{k \leq t} \sum_{j} C_{tj} \, \exp((t\Delta_{ti} - k\Delta_{ki})A_j) \, \Delta_{ki} B_{kj} \, u_i(k) . \tag{7}$$

Here, the input $u \in \mathbb{R}^d$ is a vector and the SSM is applied to each input dimension in parallel. The index $i$ is over the input dimension $d$, and $j$ indexes the hidden state dimension $n$. In the general convolution formulation we showed above, the S6 output is formulated as applying the convolution kernel $K(t, k) = C_{tj} \exp((t\Delta_{ti} - k\Delta_{ki})A_j)\Delta_{ki}B_{kj}$ on the inputs over time $u_i$. Note here that, unlike typical time-invariant SSMs, the S6 convolution kernel is unique to each $y$ as $\Delta, B, C$ change with time. Since $K(t, k) = \Big( C_{tj} \, \exp(t\Delta_{ti}A_j) \Big) \Big( \exp(-k\Delta_{ki}A_j)\Delta_{ki}B_{kj} \Big)$, the kernel is separable and we can use the procedure we introduced above to compute the S6 output in a time and memory-efficient manner (see Appendix G.1 for an efficient PyTorch Implementation of the S6).

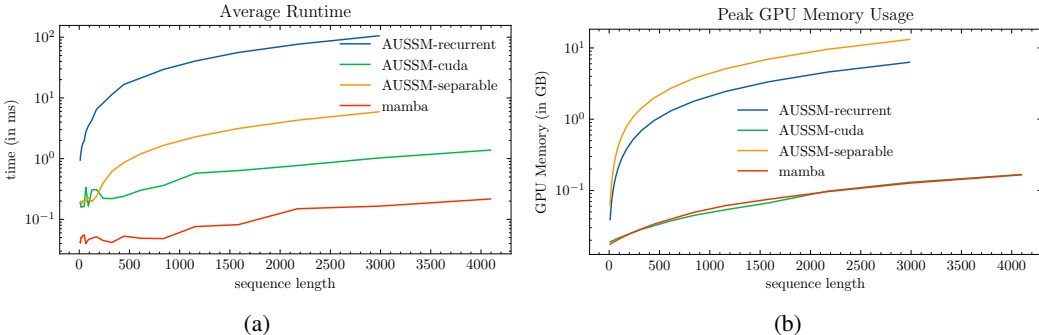

(a)                                                            (b)

Figure 2: **AUSSM with separable convolution achieves efficient runtime and memory scaling for fully adaptive SSMs.** The runtime and peak memory usage of four implementations are compared: recurrent PyTorch AUSSM, separable PyTorch AUSSM, our optimized CUDA AUSSM kernel, and the Mamba CUDA kernel. (a) The AUSSM CUDA implementation outperforms both PyTorch baselines in speed and memory efficiency, and approaches the memory efficiency of Mamba despite AUSSM's full adaptive recurrence. Notably, the PyTorch implementation of the separable convolution has better runtime efficiency compared to the recurrent implementation, albeit at a higher memory cost. (b) The AUSSM CUDA kernel has a significantly lower memory footprint, identical to that of the partially adaptive and optimized Mamba CUDA kernel.

**Separable Convolution Formulation of AUSSM:** In the case of S6, the separable formulation was easily revealed directly from the convolution form. In the case of AUSSM, this separability is not possible for the most general case. However, when the set of recurrent matrices $A_t$ is simultaneously diagonalizable, the output of the AUSSM in Equation 4 can be formulated as (See Appendix C for details)

$$y_{ti} = \Re \left[ \sum_{k \leq t} \sum_j C_j \exp\left( \mathbf{i} \sum_{l \leq t} \theta_{A_{lij}} \right) \frac{\Delta_{ki} B_j}{\exp\left( \mathbf{i} \sum_{l \leq k} \theta_{A_{lij}} \right)} u_i(k) \right] . \tag{8}$$

Here, $\theta_{A_{lij}} = \sum_r x_{ijr} u_r(k) + x_{ij}^{\text{bias}}$ is the angle argument of the unitary discretized $A'$ matrix in the polar form. Note here that as the AUSSM has complex eigenvalues, the final output is also complex, and we use only the real part of the output with the function $\Re[\cdot]$. With this formulation of the AUSSM recurrence, a memory and time efficient computation of the adaptive SSM is obtained, however, implementing this convolution directly in PyTorch can still result in high memory usage as the constant in the $O(L)$ is $bdn$ where $b$ is the batch size, $d$ is the input dimension and $n$ is the hidden dimension resulting in a large constant factor. We therefore create a CUDA kernel, where this additional complexity is hidden and the hidden state is only partially materialized in the CUDA kernel (Appendix G.2).

Another approach to improving the performance of SSMs is tensor core optimization. In tensor core optimization, special hardware features in NVIDIA GPUs called tensor cores are used to speed up matrix computations inherent in SSM implementations. This approach is not an entirely new algorithm with improved scaling behavior, but an implementation approach that enables speed-up in the special case of GPU architectures where tensor cores are available - which is most high-end GPUs available in the market. Experimental evaluations have also shown that tensor core optimization approaches can provide a constant factor increase in performance on high-end GPU hardware, but retain the same scaling behavior - the big-O scaling factor. Recent works have utilized this approach to improve the performance of time-varying SSMs, but side-step the fundamental algorithmic limitations of the problem. In contrast, our proposed algorithm for adaptive SSMs can be applied in more general cases and still provide guaranteed algorithmic scaling behavior even in GPUs where tensor core optimization is not available - for example, edge computing, other GPU makers.

## 5   Experiments

We empirically validate the theoretical claims of AUSSM by evaluating both its computational efficiency and expressivity. First, we benchmark runtime and memory usage across four implementations,

| | Task | Mamba Complex | Mamba [-1,1] | xLSTM | Mamba | AUSSM | AUSSM Hybrid |
|---|---|---|---|---|---|---|---|
| C.S | repetition | 0.09 | 0.10 | 0.09 | $0.15^3$ | $\underline{0.1993^2}$ | $\mathbf{0.45}^1$ |
| | bucket sort | 0.21 | $\underline{0.91^2}$ | 0.7 | 0.69 | $\mathbf{0.92}^1$ | $0.83^3$ |
| | majority count | 0.19 | 0.31 | $\mathbf{0.5}^1$ | $\underline{0.45^2}$ | 0.096 | $0.37^3$ |
| | majority | 0.13 | $0.63^3$ | $\underline{0.64^2}$ | $\mathbf{0.69}^1$ | 0.57 | $\underline{0.64^2}$ |
| D.C.F | solve equation | $\mathbf{0.43}^1$ | $\underline{0.24^2}$ | $\underline{0.24^2}$ | 0.05 | $0.07^3$ | $0.07^3$ |
| | mod arith | 0.12 | 0.116 | $\underline{0.15^2}$ | 0.04 | $0.13^3$ | $\mathbf{0.23}^1$ |
| Reg. | mod arith wo bra | 0.23 | 0.24 | $\mathbf{1.0}^1$ | 0.13 | $0.48^3$ | $\underline{0.53^2}$ |
| | cycle nav | 0.42 | $\underline{0.91^2}$ | 0.8 | 0.86 | $\mathbf{1.0}^1$ | $\mathbf{1.0}^1$ |
| | parity | 0.27 | $\mathbf{1.0}^1$ | $\mathbf{1.0}^1$ | $\underline{0.13^2}$ | $\mathbf{1.0}^1$ | $\mathbf{1.0}^1$ |

Table 1: **AUSSM and hybrid AUSSM+Mamba models outperform Mamba on tasks requiring counting and structured memory.** We evaluate xLSTM, Mamba, AUSSM, and a hybrid AUSSM+Mamba model on a suite of algorithmic reasoning tasks. The table shows the scaled test accuracies on each task. The tasks are grouped by their position in the Chomsky hierarchy (C.S: context-sensitive, D.C.F: Deterministic Context Free, Reg. Regular). AUSSM achieves perfect accuracy on tasks like `parity` and `cycle navigation`, which require modulo counting, validating its theoretical expressivity. While Mamba performs better on tasks such as `majority count`, the hybrid model consistently achieves the best or near-best performance across most tasks, demonstrating that combining adaptive unitary dynamics with real-valued recurrence yields a more expressive and general-purpose architecture. The scaled accuracies for xLSTM and Mamba are obtained from [31].

including our CUDA-optimized AUSSM and Mamba. Second, we assess expressivity on a suite of algorithmic tasks requiring formal language recognition, such as parity and modular arithmetic. Finally, we evaluate real-world applicability on long-sequence classification and regression tasks, demonstrating that the improved expressivity of AUSSM translates to practical performance gains.

## 5.1 Scalability Evaluation

We benchmark four implementations of AUSSM to assess efficiency: (1) a naive PyTorch recurrent version, (2) a PyTorch version using separable convolutions with a higher constant factor in the linear scaling, (3) our custom CUDA kernel, and (4) the Mamba CUDA kernel as a baseline. Experiments were run on a single NVIDIA 2080 Ti GPU with 11 GB VRAM. As shown in Figure 2, the CUDA-based AUSSM achieves significantly lower memory usage and faster inference compared to the PyTorch variants, approaching the efficiency of Mamba despite full adaptivity. The separable PyTorch implementation improves runtime over the recurrent baseline but incurs higher memory costs. Overall, our separable formulation paired with a low-level CUDA kernel enables AUSSM to scale to long sequences efficiently, validating the theoretical benefits of scalability.

### 5.1.1 Expressivity Evaluation

To evaluate formal expressivity, we benchmark AUSSM, Mamba, xLSTM [31], and a hybrid AUSSM+Mamba model on a suite of algorithmic tasks drawn from various levels of the Chomsky hierarchy. These include tasks requiring counting (e.g., parity, modular arithmetic), memory manipulation (e.g., repetition), and symbolic reasoning (e.g., equation solving). Models are trained on sequences up to length 40 and tested on lengths up to 256 to assess length generalization performance. We evaluate all the models using scaled validation accuracies to account for the differing number of output classes in the algorithmic tasks.

The results are shown in Table 1. The AUSSM achieves perfect accuracy on tasks that require modulo counting and cycle tracking, validating its theoretical ability to simulate cyclic group automata via unitary and adaptive dynamics. In contrast, Mamba fails to generalize on these tasks, consistent with the limitations of partially adaptive and dissipative models. However, AUSSM performs poorly on tasks such as majority or equation solving, where dissipative dynamics may be required for stability and information aggregation. Notably, the AUSSM hybrid model performs significantly better than

| Dataset | Heartbeat | SCP1 | SCP2 | Ethanol | Motor | Worms | Avg |
|---|---|---|---|---|---|---|---|
| Seq. len. | 405 | 896 | 1152 | 1751 | 3000 | 17984 | |
| # classes | 2 | 2 | 2 | 4 | 2 | 5 | |
| S5 | $47.8 \pm 3.1$ | $74.2 \pm 2.1$ | $10.2 \pm 3.3$ | $0.8 \pm 3.5$ | $6.0 \pm 3.9$ | $79.9 \pm 4.1$ | 36.4 |
| S6 | $\mathbf{53.0 \pm 8.3}$[1] | $65.6 \pm 2.7$ | $-0.2 \pm 9.4$ | $1.9 \pm 6.4$ | $2.6 \pm 4.7$ | $81.3 \pm 6.2$ | 34.0 |
| linoss | $51.6 \pm 3.7$ | $\mathbf{75.6 \pm 2.6}$[1] | $\mathbf{17.8 \pm 8.1}$[1] | $\mathbf{6.5 \pm 0.6}$[1] | $\mathbf{20.0 \pm 7.5}$[1] | $\mathbf{93.8 \pm 4.4}$[1] | $\mathbf{44.2}$[1] |
| Mamba | $52.4 \pm 3.8$ | $61.4 \pm 1.4$ | $-3.6 \pm 3.9$ | $3.9 \pm 4.5$ | $-4.6 \pm 4.5$ | $63.6 \pm 15.8$ | 28.8 |
| Hybrid | $\mathbf{53.0 \pm 3.8}$[1] | $64.2 \pm 4.9$ | $4.2 \pm 6.8$ | $4.7 \pm 4.1$ | $2.6 \pm 5.5$ | $82.6 \pm 3.4$ | 35.2 |

Table 2: **Hybrid AUSSM with Mamba achieves competent performance on long time-series classification benchmarks.** We evaluate the hybrid model on six UEA datasets spanning a wide range of sequence lengths and domains. The table shows the scaled test accuracies for the different models compared to the hybrid AUSSM. The hybrid AUSSM consistently outperforms the base Mamba and achieves competent accuracy across datasets. These results demonstrate that the increased expressivity of AUSSM, when combined with Mamba's stability, translates into strong real-world performance even on long and complex sequence data. Our model is evaluated on a statistically rigorous test with 20 different seeds to obtain a better estimate of test accuracy to reduce the reliance on arbitrary evaluation seeds used in prior works [20].

all existing RNNs, including the xLSTM, suggesting that AUSSMs and Mamba blocks are synergistic and exhibit performance benefits that neither individual model provides. These results empirically support our theoretical claim that hybrid models combining AUSSM and Mamba maximize the expressivity of diagonal SSMs under the Krohn–Rhodes framework.

## 5.2 Long Time-Series Classification and Regression Benchmark

To evaluate the practical benefits of our architecture, we test the hybrid AUSSM+Mamba model on a suite of UEA long-time-series classification benchmarks [32] and the challenging Weather regression benchmark. We take the AUSSM block as a drop-in replacement for an existing Mamba backbone. Specifically, we randomly selected a fixed number of Mamba blocks in a deep Mamba SSM model and replaced them with the AUSSM blocks. The UEA tasks feature much longer sequences than the algorithmic benchmark, with lengths ranging from 405 in the `Heartbeat` dataset to over 17,000 in the `Worms` dataset. For regression, we use the challenging `Weather` dataset where climate variables are forecasted 720 steps into the future, given a window of 720 timesteps. These benchmarks present a more realistic and diverse set of challenges, which includes physiological signals, chemical concentrations, motion data, and climate, each requiring the model to capture both local and global temporal dependencies.

For the UEA benchmarks and the weather dataset, we used identical hyperparameter strategies to those used by the models we compare against. During testing, we found that previous works used five randomly chosen seeds to evaluate the test performance. This is not easily reproducible, as the particular choice of the seeds influences the specific test datasets that are chosen for evaluation and may produce biased results. We instead use a statistically rigorous technique where the best hyperparameter model is chosen based on the validation set performance on five random seeds, and the test accuracy is evaluated on random train-test splits on the selected model with 20 different seeds to produce better test accuracy estimates. We scaled test accuracies with the baseline and report the results in Table 2. The hybrid AUSSM+Mamba model achieves substantial improvements over the partially adaptive Mamba SSM on average, even under the modified testing protocol. The results demonstrate that the improved expressivity of AUSSM carries over to real-world tasks when appropriately combined with the stability and inductive biases of partially adaptive models. Notably, the hybrid model achieves these results while maintaining high efficiency: all experiments except `EigenWorms` were run on a single NVIDIA 2080 Ti GPU with 11 GB of VRAM, in contrast to the large-scale hardware (e.g., 100 GB A100 GPUs) typically used for long-sequence modeling. The `Eigenworms` dataset was trained on an L4 GPU with 23 GB VRAM due to its larger size.

| Model | Mean Absolute Error ↓ |
|---|---|
| Informer | 0.731 |
| LogTrans | 0.773 |
| LSTMa | 1.109 |
| LSTnet | 0.757 |
| S4 | 0.578 |
| LinOSS | 0.508[3] |
| Mamba | 0.464[2] |
| AUSSM Hybrid | **0.342**[1] |

Table 3: **Hybrid AUSSM+Mamba achieves state-of-the-art performance on long time-series weather forecasting benchmark.** We evaluate AUSSM against 7 different models on the challenging weather forecasting benchmark, where climate variables are forecasted 720 timesteps into the future. AUSSM Hybrid achieves state-of-the-art performance on the task, improving on the base Mamba model and all the other models.

## 6   Discussion

In this work, we address the expressivity-scalability tradeoff in state space modeling. Existing SSMs like Mamba are scalable but limited in expressivity due to fixed or partially adaptive recurrence. On the other hand, more general LTV SSMs are more expressive but do not have an efficient and scalable parallel implementation. We introduce the Adaptive Unitary State Space Model (AUSSM), which uses input-dependent skew-symmetric recurrence to achieve both unitary evolution and high expressivity. We showed that theoretically, AUSSM can implement modulo counters and simulate a broad class of regular languages, maximizing expressivity among diagonal SSMs when combined with Mamba under the Krohn–Rhodes framework. To ensure scalability, we develop a separable convolution formulation and a custom CUDA kernel, enabling linear-time training despite full adaptivity. Experimental analysis on standard benchmark tasks showed that AUSSM achieves strong performance on symbolic reasoning tasks and serves as an effective drop-in enhancement to Mamba for long-range sequence modeling. Together, these results suggest that adaptive unitary recurrence is a powerful inductive bias for both symbolic and continuous sequence tasks.

**Limitations.** One limitation related to expressivity is that AUSSM is capable of LTV recurrence only through its linearly adaptive (input-dependent) recurrent matrix. For the more general LTV recurrence, the recurrent matrix needs to have the capability for non-linear input-dependence that is additionally dependent on time.

Further, our separable kernel approach used for optimization relies on the assumption that recurrent matrices are simultaneously diagonalizable, limiting the ability to express languages beyond solvable regular languages. While the separable kernel has identical scaling behavior to efficient LTI models, it is still a constant factor higher. Hybrid AUSSM+Mamba models show promise, but the best strategy for combining blocks is not yet well understood. Another limitation in the parallel scan-based algorithm we proposed is that, in the case of high-end NVIDIA GPUs, alternate tensor core approaches may provide even better absolute speedup, although with the same scaling behavior. An alternate tensor core-based algorithm for AUSSM is an interesting avenue for future work in real-world applications. Finally, due to resource limitations, our evaluations are limited to modest-scale tasks; further validation on foundation-model scale benchmarks is needed.

## 7   Acknowledgements

We thank Reda Boumasmoud for his input and suggestions on an earlier draft of this manuscript. We also thank Michael Hahn for a useful conversation about SSM expressivity. T.A.K. acknowledges the Kempner Institute for the Study of Natural and Artificial Intelligence at Harvard University for funding during work on this article. T.J.S. acknowledges funding from ONR N00014-23-1-2069. A.K. and H.T.S. acknowledge NSF for EAGER: Neural Networks that Temporally Change (NOTCH).

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

# A    Related Work

In this appendix section, we provide an extensive discussion of related work on unitary RNNs, conserved dynamics in the brain, and the computational complexity of RNNs and SSMs.

## A.1    Unitary RNNs

Recurrent neural networks (RNNs) are powerful models for processing sequential data, but their training is often hindered by the vanishing and exploding gradient problem, which limits their ability to capture long-range dependencies [13]. A major source of this instability is the repeated multiplication of the hidden state by the recurrent weight matrix, which causes gradients to exponentially decay or grow depending on the spectral norm of the matrix. To address this, a prominent line of research constrains the recurrent dynamics to be unitary, ensuring that the hidden state evolution preserves its norm through time. Such norm-preserving dynamics prevent gradient magnitude degradation and are mathematically analogous to energy-conserving, reversible dynamical systems.

The first major work in this direction was Unitary Evolution Recurrent Neural Network (uRNN) [33] that proposed parameterizing the recurrent weight matrix as a product of structured unitary matrices (including diagonal phase matrices, Fourier transforms, and Givens rotations) to ensure exact unitarity while retaining efficient computation and gradient flow. However, this parameterization limited expressivity due to its constrained structure. To overcome this, Full-Capacity Unitary RNN [34] optimized directly over the unitary group using manifold optimization techniques. By employing the Cayley transform and optimization on the Stiefel manifold, they achieved full representational power while preserving unitary constraints.

Subsequent research explored alternative approaches for maintaining orthogonality and improving trainability. Efficient Unitary Neural Networks (EUNN) [35] used parameter-efficient decompositions enabling flexible trade-offs between computational cost and expressivity. Vorontsov et al. Orthogonality regularization methods softly constrain the recurrent weight matrix toward being orthogonal rather than enforcing strict unitarity, allowing some deviation to improve learning flexibility [36]. Similarly, Mhammedi et al. [37] developed a real-valued orthogonal RNN based on Householder reflections, which guarantees orthogonality through efficient matrix parameterizations. Later, Lezcano-Casado and Martínez-Rubio [38] introduced an elegant exponential parametrization of orthogonal and unitary matrices via skew-symmetric matrices, offering a smooth and numerically stable way to maintain orthogonality during training.

A key challenge in unitary and orthogonal RNNs is the choice of suitable nonlinearities. Standard nonlinearities such as ReLU or tanh can break the norm-preserving property of the recurrent map, leading to unstable or dissipative dynamics. To address this, [33] introduced modReLU, a complex-valued nonlinearity that preserves the phase of hidden activations while applying a learned threshold on their magnitudes. Subsequent studies explored alternatives such as zReLU [39], scaled tanh, and phase-preserving nonlinearities for complex-valued networks. In [40], the authors analyzed the interplay between orthogonality, nonlinearity, and gradient flow, providing theoretical insights into how orthogonal constraints help maintain long-term dependencies even in nonlinear regimes. More recently, Chang et al. [41] proposed Antisymmetric RNNs, where the recurrent weight matrix is constrained to be near-skew-symmetric, thereby approximating a Hamiltonian or energy-conserving flow; this represents a bridge between strict unitarity and continuous-time neural dynamics.

The success of unitary and orthogonal RNNs has motivated several extensions. For example, Tallec and Ollivier [42] analyzed time-warping effects and timescale adaptation in RNNs, showing how orthogonality can help control effective memory timescales. Lezcano-Casado [43] further generalized the parameterization of orthogonal operators to improve optimization stability across different architectures. The insights from unitary evolution have also influenced more recent Structured State Space Models (SSMs) [9], which impose spectral stability and linear time-invariant dynamics to achieve long-range sequence modeling with recurrent efficiency. These connections underscore the broader relevance of norm-preserving and energy-conserving formulations for building stable, interpretable, and trainable dynamical models.

## A.2 Unitary Dynamics in the Brain

A growing body of experimental and theoretical work indicates that neural population activity often evolves according to dynamics that are, in important respects, conserved or weakly dissipative, with clear implications for how the brain stores and transforms information. Conserved dynamics here refers to trajectories or modes that preserve key quantities (e.g., norms, phase relationships, low-dimensional energy-like functions) over behaviorally relevant timescales, producing smooth, reversible, or rotational population flows rather than rapidly diffusive or purely dissipative responses. Empirically, such phenomena appear across modalities and brain areas: propagating and wave-like activity has been observed in sensory and motor cortices [44, 45, 46, 47], low-dimensional structured trajectories that persist across trials and conditions have been reported in motor and prefrontal populations [48, 49, 48, 50, 51], and cortical responses frequently exhibit coherent oscillatory/rotational components that suggest near-unitary evolution within a task-relevant subspace [49, 52]. These conserved modes are often embedded within a larger high-dimensional network state, but they dominate the behaviorally relevant dynamics and appear to support robust temporal computation, short-term memory, and smooth transformation between input and output representations.

Theoretical and modeling studies have proposed multiple mechanistic origins for conserved or weakly dissipative neural dynamics. Balanced excitatory–inhibitory network regimes can produce rich transient dynamics with slow decay or quasi-conserved activity on short timescales; classic balanced-network analyses show how tightly coupled excitation and inhibition enable irregular activity while constraining macroscopic statistics, and follow-up work has shown how such balance supports structured transient trajectories [53, 54]. Network architectures with antisymmetric or near-skew-symmetric connectivity produce rotational and energy-like flows that are approximately conservative; control- and dynamical-systems-oriented analyses have demonstrated that small departures from strict antisymmetry (e.g., weak damping or inputs) permit flexible routing and readout while retaining the stability advantages of norm-preserving flows [55, 56]. In parallel, reservoir- and recurrent-network modeling (including trained networks initialized in richly recurrent regimes) has shown that networks can learn to generate low-dimensional conserved trajectories that implement computations (e.g., context-dependent integration, short-term memory) with high robustness to noise and parameter changes [57, 58].

Methodologically, the identification of conserved modes in neural data relies on dimensionality-reduction and dynamical-systems tools that explicitly search for rotational, low-dimensional, or wave-like structure. Approaches range from linear subspace methods that highlight persistent modes to more specialized decompositions and dynamical fits that extract antisymmetric components, traveling-wave decompositions, or stable latent manifolds; these analyses have repeatedly revealed that a relatively small number of conserved or near-conserved modes often capture most of the task-relevant variance, even when single-neuron responses are heterogeneous [49, 50, 59]. Importantly, conserved neural dynamics appear functionally beneficial: by concentrating computation in norm-preserving subspaces, the brain can transform and transmit signals with minimal degradation, enabling temporally-extended operations such as sequence generation, motor command shaping, and transient working memory without continual external reinforcement.

Finally, the convergence of empirical findings and theoretical models has motivated viewing cortical and subcortical circuits through the lens of structured dynamical primitives—rotations, waves, and nearly-Hamiltonian flows—that are neither purely feedforward nor purely dissipative. This perspective helps explain phenomena such as reliable single-trial trajectories, robustness of latent dynamics across learning and perturbation, and the coexistence of fast irregular activity with slow conserved modes [48, 51, 54]. Recognizing conserved dynamics in the brain also provides a bridge to normative and engineering approaches (e.g., unitary or antisymmetric RNNs, energy-based network formulations) that aim to replicate the computational advantages of biological circuits while offering interpretable and stable mechanisms for long-timescale processing.

| Model family (examples) | Constant diagonal *S4* | Adaptive diagonal *Mamba, AUSSM-hybrid (ours)* | Adaptive non-diagonal *DeltaProduct, RWKV-7* |
|---|---|---|---|
| **Adaptive** | no | yes | yes |
| **Diagonal** | yes | yes | no |
| **Eigenvalues** | [0,1) | various | complex |
| **Languages** | (subset of) star-free langs. | star-free to solvable langs. | permutation group langs. |

Table 4: High-level comparison of three families of linear recurrent neural networks (LRNNs) by adaptivity and diagonality, and their confirmed expressivity under standard assumptions.

| Model | Mamba | Mamba-negative | AUSSM-hybrid (ours) | Mamba-complex |
|---|---|---|---|---|
| **Adaptive** | indirect | indirect | yes (AUSSM) | indirect |
| **Diagonal** | yes | yes | yes | yes |
| **Eigenvalues** | $(0, 1)$ | $(-1, 1)$ | unitary (AUSSM) | complex |
| **Star-free** | ✓[11, Thm. 4] | ✓[11, Thm. 4] | ✓[11, Thm. 4] | ✓[11, Thm. 4] |
| **Solvable** | ✗[11, Thm. 4] | ✗[25, Thm. 2] | ✓(Thm. 2) | ✓[11, Thm. 21] |
| **Regular** | ✗[28, Thm 4.4] | ✗[28, Thm 4.4] | ✗[28, Thm 4.4] | ✗[28, Thm 4.4] |

Table 5: More fine-grained comparison of design features and formal language expressivity of Mamba-like models.

### A.3 Computational Expressivity Theory of Linear RNN Architectures

As linear RNNs (LRNNs), of which SSMs are a special case, have gained in performance and become a viable alternative to transformers for sequence processing tasks, their formal expressive power has garnered interest. LRNN denotes the family of recurrent neural networks whose transition function is a linear or affine transformation of the hidden state (note, however, that the linear transition may be a non-linear function of the input at each time step). In contrast, traditional RNNs such as the Elman RNN [60], LSTM[61], or GRU[62] update their hidden state non-linearly between time steps. State space models (SSMs) such as S4 [9] and Mamba [10] are a subtype of LRNNs motivated by continuous linear dynamical systems that are discretized to work recurrently as LRNNs. In terms of formal expressivity, [28] and [11] point out that most SSMs make architectural decisions that limit their ability to model formal languages. The main factors impacting expressivity are:

- Adaptivity - Whether the transition matrix is held constant over time or is a (non-linear) function of the input at the current time step.

- Diagonality - Imposing that the transition recurrence is a diagonal or diagonalizable matrix (simultaneously for all inputs).

- Eigenvalue range - Restricting the transition recurrence to matrices with eigenvalues in a specific range (e.g., non-negative, real between -1 and 1, complex unitary, etc.).

As well as impacting their expressivity, these decisions also determine the scalability of the architecture, since certain restrictions allow for more efficient implementations, e.g., the product of diagonal matrices can be computed more efficiently than that of dense matrices. There appears to be a distinct tradeoff between expressivity and scalability, informally dubbed the parallelism tradeoff [63]. See Tab. 4 for a comparative high-level overview of different families of LRNNs and their expressivity and relative scalability.

**Adaptivity** Some of the earlier SSM variants, such as S4, were non-adaptive (or time-invariant), making them very scalable through the use of convolution over the whole sequence using a pre-computable constant convolution kernel. As [28] points out, input-independence ensures that the expressivity of SSMs is upper-bounded by the circuit complexity class TC0, while some regular languages require circuit complexity NC1 (it is widely assumed that TC0 != NC1). More recent SSM architectures add adaptivity through various means, e.g., the Mamba recurrence is indirectly

input-dependent via its time discretization factor $\Delta$. Similarly, LRNNs such as RWKV-7 [64] or DeltaProduct [65], and non-linear RNNs such as xLSTM [31], are adaptive through mechanisms such as input-dependent gating factors. The transitions of our AUSSM component are directly input-dependent, making our architecture fully adaptive (see Tab. 5).

**Diagonality**    Another critical factor is whether the transition recurrence is diagonal or simultaneously diagonalizable for different inputs. [11, Thm. 21] shows that such diagonal SSMs can recognize solvable languages, but [28] again shows that such SSMs are contained within circuit complexity class TC0, meaning they cannot recognize all regular languages, assuming TC0 != NC1. The upshot is that diagonal transition matrices mean quicker or less memory-intensive computation for training and inference, making it a very attractive architectural decision as employed by S4, Mamba, and others. For this reason, we also choose to keep diagonality for our AUSSM and hybrid architecture, and mainly compare performance between diagonal SSMs.

In contrast, the design of DeltaProduct or RWKV-7 attempts to create non-diagonal LRNN architectures whose transition matrices are products of generalized householder matrices, which are diagonal matrices plus an added rank-1 component. Such models can recognize all regular languages [25, 64], albeit at the price of additional time and memory cost.

We also compare the performance of our architecture to that of xLSTM, which, as an extension of the traditional LSTM, is an RNN but not an LRNN since the recurrence is non-linear in the hidden state. Since LSTMs can recognize more than just regular languages [66], we use this as an upper-bound comparison to a stronger model. In fact, while the authors do not formally prove that xLSTMs can recognize all regular languages, the experimental results show strong performance on this class of languages.

**Eigenvalue range**    Within the realm of adaptive diagonal (or diagonalizable) SSMs in particular, the eigenvalue range of the transition matrices plays a crucial role. This is because, as [11, Thm. 4] proves, non-negative eigenvalues restrict diagonal SSMs to the class of star-free regular languages, while negative eigenvalues allow for the recognition of non-star-free languages such as parity. [25] point out that Mamba can be trivially adapted to have negative eigenvalues without additional computational cost. However, negative eigenvalues alone are not enough to recognize all solvable languages; complex eigenvalues are required, e.g., for solvable languages like $\mod n$ parity for $n > 2$ [25, Thm. 2]. In non-diagonal LRNNs, multiplying generalized Householder matrices as in DeltaProduct or RWKV-7 can yield eigenvalues with non-zero imaginary components, raising their expressivity to include all solvable languages (and, indeed, all regular languages [65]).

In order to recognize all solvable languages with diagonal SSMs, we need to extend the eigenvalue range to complex numbers. The simplest way to do this is to just use Mamba with complex hidden states. This incurs additional overhead, however, because it increases the parameter count to include all possible complex values. As we show in §E, in order to accept all solvable languages, we only need to add SSM components with unitary complex eigenvalues, which is why we introduce AUSSM components to Mamba, allowing the hybrid architecture to model all solvable languages with minimal overhead. Additionally, while formally Mamba with complex values is just as expressive as our architecture, our experiments showed that Mamba with complex eigenvalues fails to learn even simple formal tasks that our hybrid architecture can perform with perfect accuracy, indicating that the additional restriction to unitary values is indeed helpful for learning formal languages. See Tab. 5 for a comparison of Mamba-like models with their relative advantages and disadvantages.

## B    Limitations of Non-Adaptive/Partially Adaptive SSMs

The expressivity of different classes of SSMs is defined by the types of dynamical systems and formal languages they are able to simulate. Appendix E analyzes the formal language expressivity and limitations of different classes of SSMs. In this section, we analyze expressivity related to different kinds of dynamical systems. First, we show that in line with Figure 1, SSM expressivity can be arranged in the order `LTI real spectra` $\subset$ `LTI complex` $\subset$ `LTV partial` $\subset$ `LTV`. The models that have higher expressivity can simulate the models lower in the expressivity scale. Since `LTV w unitary spectra` cannot be arranged precisely in this scale, we show an example of a class of multitimescale processes that a partial LTV model like Mamba cannot simulate in a fixed hidden state and layer limits.

**Expressivity of Single Block SSMs**   The dynamical systems that can be simulated by single block SSMs without non-linearities can be arranged in the order LTI real spectra $\subset$ LTI complex $\subset$ LTV partial $\subset$ LTV.

*Proof*: For the proof, we start with the most general LTV SSM and show that the next lower class SSM is a special case. We do the same for all the subsequent SSM classes in the expressivity chain.

The single block LTV SSM has the following discrete form:

$$\begin{cases} \frac{\mathrm{d}x(t)}{\mathrm{d}t} = \exp(\Delta_t\, A_t)\ x(t) + \Delta_t B_t\, u(t)\,, \\ y(t) = C_t\, x(t)\,. \end{cases}$$

The next lower class of ssm: LTV partial has the following form

$$\begin{cases} \frac{\mathrm{d}x(t)}{\mathrm{d}t} = \exp(\Delta_t A)\ x(t) + \Delta_t B_t\, u(t)\,, \\ y(t) = C_t\, x(t)\,. \end{cases}$$

Note here that the $\Delta_t$ is a scalar that varies with time, but $A$ is a fixed matrix. This can be derived as an instance of the LTV with $A_t = \Delta_t A$ where the equivalence between the two holds only in the case where the dimensionality of the SSM is 1. Similarly, the next lower class LTI complex has the following form

$$\begin{cases} \frac{\mathrm{d}x(t)}{\mathrm{d}t} = \exp(\Delta A)\ x(t) + \Delta B\, u(t)\,, \\ y(t) = C\, x(t)\,. \end{cases}$$

This is an instance of the LTV partial with $\Delta_t = \Delta$, $B_t = B$ and $C_t = C$, which means all the matrices are time invariant. The final class LTI real spectra is an instance of LTI Complex where the eigenvalues are further restricted to have 0 angle in the imaginary plane.

LTV is the most general class, but it is computationally infeasible to simulate the most general case. The non-diagonalizability of general matrix classes requires performing a full $O(n^3)$ matrix computation at each time step. Hence LTV w unitary spectra with simultaneously diagonalizable unitary matrices is chosen as a principled middle ground. It is, however, not instantly apparent how LTV w unitary spectra compares against LTV partial. To illustrate the difference, we introduce an example of a multi-timescale process.

**Multi-timescale features**: A time-series $u(t) \in \mathbb{R}$ is said to have multi-timescale features if the hidden state can be factorized into the following form:[6]

$$x(t+1) = \begin{pmatrix} f(t) & 0 \\ 0 & g(t) \end{pmatrix} x(t)\,.$$

Where $f(t) \in \mathbb{C}, g(t) \in \mathbb{C}$ are general complex-valued time-varying functions and $f(t) \neq cg(t)$ for some constant $c$. That is, the timeseries exhibits at least two *independent* features denoting two different timescales.

**partial LTV SSMs in multi-timescale timeseries**: partial LTV SSMs are not able to represent multi-timescale features in data.

*Proof*: The proof is by contradiction. If partial LTV SSMs are able to solve multi-timescale timeseries, the following $\begin{pmatrix} f(t) & 0 \\ 0 & g(t) \end{pmatrix} = \Delta_t A$ is true. Solving the system for $A$ leads to a constraint on $f(t) = cg(t)$ where $c$ is some constant. This is true only when one of the functions is a constant multiple of the other, that is *the two functions are dependent and have the same timescale (with a possible constant factor difference)*.

**LTV w unitary spectra SSMs in multi-timescale timeseries**: LTV w unitary spectra SSMs can represent multi-timescale features in data as long as the $f(t), g(t) \in \exp(i\theta)$ where $\theta \in [-\pi, \pi]$. $f(t), g(t)$ can be independent.

*Proof*: We first substitute $f(t) = \exp\big(\mathbf{i}\,\theta^f(t)\big)$ and $g(t) = \exp(\mathbf{i}\,\theta^g(t))$. The resulting dynamical system has $f(t)$ and $g(t)$ as eigenvalues, which have unit magnitude themselves. This is the definition of LTV w unitary spectra.

---

[6]The results trivially extend to systems with more than two dimensions

To summarize, if $f$ and $g$ are independent (e.g., $f(t) = t^2$, $g(t) = t$), then the partial LTV system cannot represent the multi-timescale features in the hidden state. On the other hand, `LTV w unitary spectra` imposes a weaker constraint where the only requirement is that $f(t)$ and $g(t)$ are constrained to the unit circle in the imaginary plane; the timescales of the two variables can be independent.

**Note 1.** *In the main text, when we say that AUSSM is a diagonal LTV system, we mean that AUSSM is capable of LTV recurrence through its adaptive (input-dependent) recurrent matrix. For the more general LTV recurrence, the recurrent matrix needs to have the capability for non-linear dependence which AUSSM currently does not support.*

**Note 2.** *In this section, we showed that a single AUSSM block with a fixed model dimension and hidden state size can represent functions that Mamba cannot represent with the same hyperparameters (it may need a greater width or more layers for the same function). At first glance, this seems to contradict the results shown in Tab. 5, which posit that Mamba with complex entries is as expressive as our hybrid architecture. The explanation is that in the formal language expressivity analysis in §3.1 and §E is concerned with the expressivity of the* whole architecture class over any finite parametrization*, rather than a specific model parametrization. The two analyses, therefore, keep different quantities constant: the expressivity analysis is about the existence of any finite instantiation of the model class that realizes a given language, while the fixed-hyperparameter single-layer measures relative capacity at constant size.*

## C AUSSM Derivation

We derive the AUSSM from a controlled and adaptive version of the skew-symmetric ODE used in the jPCA procedure in computational neuroscience, given below. The skew-symmetric ODE is first discretized using the Zero Order Hold procedure and then parameterized in polar coordinates. The steps to obtain the final AUSSM formulation are provided below.

$$\begin{cases} \frac{dx(t)}{dt} = A_t \, x(t) + B \, u(t) \,, \\ y(t) = C \, x(t) \,. \end{cases} \tag{9}$$

The above ODE is discretized following the Zero Order Hold procedure with a step size of $\Delta_t$ (note that the step size is also time varying like the recurrent matrix)

$$\begin{cases} x(t) = \exp(\Delta_t A_t) \, x(t-1) + \Delta_t B \, u(t) \,, \\ y(t) = C \, x(t) \,. \end{cases} \tag{10}$$

The convolution form of the above system can be derived from this recurrence as shown below (assuming $x(0) = 0$)

$$y(1) = C\Delta_1 CBu(1) \tag{11}$$
$$y(2) = C \exp(\Delta_2 A_2)\Delta_1 Bu(1) + \Delta_2 CBu(2) \tag{12}$$
$$\vdots \tag{13}$$
$$y(t) = C \sum_{k=1}^{t-1} \left( \prod_{l=k+1}^{t} \exp(\Delta_l A_l) \right) \Delta_k Bu(k) + \Delta_t CBu(t) \tag{14}$$

Note that without additional assumptions on $A$, the matrix exponential and the repeated products cannot be simplified further, which can result in computationally inefficient approaches to compute the output. We draw motivation from the use of structured matrices in efficient SSM implementations and propose that $A_t$ belongs to a class of matrices that are simultaneously diagonalizable with the same basis. Let this diagonalizable basis be $P$.

$$y(t) = C \sum_{k=2}^{t-1} P \left( \prod_{l=k+1}^{t-1} \exp(\Delta_l \Lambda(A_l)) \right) P^{-1} \Delta_k Bu(k) + \Delta_t CBu(t) \,, \tag{15}$$

where $\Lambda(A_l)$ is the diagonal matrix with the eigenvalues of $A_l$ on the diagonal. Now, the repeated matrix product has a simplified form as shown below.

$$y(t) = CP \sum_{k=2}^{t-1} \left( \exp \left( \sum_{l=k+1}^{t-1} \Delta_l \, \Lambda(A_l) \right) \right) P^{-1} \Delta_k Bu(k) + \Delta_t CBu(t) \,, \tag{16}$$

For a new set of $B'$ and $C'$ such that $C' = CP$ and $B' = P^{-1}B$, we get

$$y(t) = C' \sum_{k=2}^{t-1} \left( \exp\left( \sum_{l=k+1}^{t-1} \Delta_l \, \Lambda(A_l) \right) \right) \Delta_k B' u(k) + \Delta_t C' \, B' \, u(t) \,, \tag{17}$$

The above equation undergoes one additional simplification, which reveals the unitarity of the discrete dynamical system. Since $A_l$ is a skew-symmetric matrix, the eigenvalues $\Lambda(A_l)$ are purely imaginary, meaning the above equation simplifies further in the polar form of $A_l$.

$$y(t) = C' \sum_{k=2}^{t-1} \left( \exp\left( \mathbf{i} \sum_{l=k+1}^{t-1} \Delta_l \, \Im(\Lambda(A_l)) \right) \right) \Delta_k B' u(k) + \Delta_t C' \, B' \, u(t) \,, \tag{18}$$

where $\mathbf{i}^2 = -1$ is the complex iota and $\Im(.)$ is the function that obtains the imaginary component of a complex number. Since $C'$ and $B'$ are also complex due to the multiplication with $P$, we use polar forms for them too to finally obtain

$$y(t) = R_C \exp(\mathbf{i}\,\theta_C) \sum_{k=2}^{t-1} \left( \exp\left( \mathbf{i} \sum_{l=k+1}^{t-1} \Delta_l \, \Im(\Lambda(A_l)) \right) \right) \Delta_k R_B \exp(\mathbf{i}\theta_B) u(k) \tag{19}$$
$$+ \Delta_t R_C \exp(\mathbf{i}\,\theta_C) \, R_B \exp(\mathbf{i}\,\theta_B) \, u(t) \,.$$

To handle a $d$-dimensional input, this formulation is replicated $d$ times for each input dimension. For adaptivity, we use where $\Lambda(\Delta_l A_l)_j = \sum_r \chi_{jr} \, u_r(l) + \chi_j^{\text{bias}}$ and $\Delta_{lj} = \sum_r \chi_{jr}^{\Delta} \, u_r(l) + \chi_j^{\Delta \text{bias}}$, We use the above formulation in our experiments and parameterize the following for learning: $R_C, \theta_C, R_B, \theta_B, \chi_{jr}, \chi_j^{\text{bias}}, \chi_j^{\Delta \text{bias}}, \chi_{jr}^{\Delta}$.

## D   Eigenvalue Analysis

**Lemma 3** (Exponential of a Skew-Symmetric Matrix is Orthogonal). *Let $A \in \mathbb{R}^{n \times n}$ be a real skew-symmetric matrix, i.e., $A^\top = -A$. Then the matrix exponential $\exp(\Delta A)$ is orthogonal for any $\Delta \in \mathbb{R}$, i.e.,*
$$\exp(\Delta A)^\top \exp(\Delta A) = I.$$

*Proof.* Let $U = \exp(\Delta A)$. Then,
$$U^\top = (\exp(\Delta A))^\top = \exp(\Delta A^\top) = \exp(-\Delta A),$$
since $A^\top = -A$. Therefore,
$$U^\top U = \exp(-\Delta A) \exp(\Delta A) = \exp(0) = I,$$
which shows that $U$ is orthogonal. $\qquad\square$

**Lemma 4** (Marginal Stability of Discrete-Time Dynamics). *Let $A \in \mathbb{R}^{n \times n}$ be a real skew-symmetric matrix and define $\Phi = \exp(\Delta A)$ for some $\Delta > 0$. Then all eigenvalues of $\Phi$ lie on the complex unit circle. In particular, the discrete-time linear system*
$$x(t) = \Phi x(t-1)$$
*is marginally stable.*

*Proof.* The eigenvalues of a real skew-symmetric matrix $A$ are purely imaginary, i.e., $\lambda_j = i\omega_j \in i\mathbb{R}$. The eigenvalues of $\Phi = \exp(\Delta A)$ are then
$$\mu_j = \exp(\Delta \lambda_j) = \exp(i\Delta \omega_j),$$
which all lie on the complex unit circle since $|\exp(i\theta)| = 1$ for all $\theta \in \mathbb{R}$. Hence, the system exhibits marginal stability. $\qquad\square$

**Lemma 5** (Norm Preservation under Skew-Symmetric Dynamics). *Let $A \in \mathbb{R}^{n \times n}$ be a real skew-symmetric matrix, and let $\Phi = \exp(\Delta A)$. Then for any $x \in \mathbb{R}^n$,*

$$\|\Phi x\|_2 = \|x\|_2.$$

*Hence, the transformation does not amplify or diminish the norm of the state vector, preventing both gradient explosion and vanishing during backpropagation through time.*

*Proof.* Since $\Phi$ is orthogonal by Lemma 1, we have:

$$\|\Phi x\|_2^2 = (\Phi x)^\top (\Phi x) = x^\top \Phi^\top \Phi x = x^\top x = \|x\|_2^2.$$

Taking the square root yields $\|\Phi x\|_2 = \|x\|_2$. $\qquad\qquad\qquad\qquad\qquad\qquad \square$

**Lemma 6** (Input-Modulated Rotation Frequencies via Skew-Symmetric Generator). *Let $A : \mathbb{R} \to \mathbb{R}^{n \times n}$ be a smooth function such that $A(u)$ is skew-symmetric for all $u \in \mathbb{R}$. Then for each $u \in \mathbb{R}$, all eigenvalues of $A(u)$ lie on the imaginary axis, and the eigenvalues of the discrete-time transition matrix $\Phi(u) = \exp(\Delta A(u))$ lie on the complex unit circle.*

*Furthermore, the eigenvalues of $A(u)$ depend continuously on $u$, and thus the angular frequency of state-space rotation is smoothly and directly modulated by the input.*

*Proof.* Let $A(u) \in \mathbb{R}^{n \times n}$ be skew-symmetric for all $u \in \mathbb{R}$, i.e., $A(u)^\top = -A(u)$. It is a well-known result from linear algebra that real skew-symmetric matrices have purely imaginary eigenvalues or zero.

Let $\lambda_j(u) \in \mathbb{C}$ be an eigenvalue of $A(u)$. Since $A(u)$ is real and skew-symmetric, $\lambda_j(u) = i\omega_j(u)$ for some $\omega_j(u) \in \mathbb{R}$, and the eigenvalues come in complex-conjugate pairs if nonzero.

Now, consider the discrete-time transition matrix:

$$\Phi(u) := \exp(\Delta A(u)).$$

Because the exponential of a skew-symmetric matrix is orthogonal (by Lemma 1), $\Phi(u)$ is an orthogonal matrix. The eigenvalues of an orthogonal matrix with determinant 1 lie on the complex unit circle, i.e.,

$$|\mu_j(u)| = 1 \quad \text{for all eigenvalues } \mu_j(u) \text{ of } \Phi(u).$$

Furthermore, the eigenvalues of $\Phi(u)$ are given by

$$\mu_j(u) = \exp(\Delta \lambda_j(u)) = \exp(i \Delta \omega_j(u)),$$

so their arguments (i.e., angular velocities) are precisely modulated by the real-valued frequencies $\omega_j(u)$, which in turn depend on the input $u$.

To show that the rotational frequencies vary continuously with $u$, recall that the eigenvalues of a smooth matrix function $A(u)$ depend continuously on $u$, provided that $A(u)$ has distinct eigenvalues or that perturbations are small (which holds generically due to the structure of skew-symmetric matrices). Since $A(u)$ is assumed to be smooth, all $\omega_j(u)$ vary continuously with $u$, and therefore so do the corresponding angles $\Delta \omega_j(u)$ of the discrete-time rotation matrix.

$$\qquad\qquad\qquad\qquad\qquad\qquad\qquad\qquad\qquad\qquad\qquad\qquad\qquad \square$$

# E   Formal Language Expressivity

Our formal expressivity analysis uses the setting and proofs of [11] as a starting point. That is, we abstract away architectural details without loss of generality, and directly work with the already discretized form of the SSM. We assume floating-point arithmetic where the precision is logarithmically bounded in the sequence length, i.e., at most $\mathcal{O}(\log n)$ bits of precision on inputs of length $n$. Here, we briefly reiterate a somewhat abstract definition of our SSM to simplify the expressivity proofs.

**Definition 1** (SSM layer). *A single SSM layer is a sequence-to-sequence map $\mathbb{R}^d \to \mathbb{R}^d$, $(u_t) \mapsto (y_t)$ for $t \in [T]$ for sequence length $T$. It is defined recurrently by*

$$x_t = A_t \circ x_{t-1} + B_t \tag{20}$$

*where $\circ$ is elementwise multiplication, $x_0 \in \mathbb{C}^m$ with $m = n \cdot d$, and $A, B \colon \mathbb{R}^d \to \mathbb{C}^m$ are smooth, input-dependent maps with $A_t = A(u_t)$ and $B_t = B(u_t)$. Note that $A$ already subsumes the discretization variable $\Delta$, which is itself a function of the input, as introduced in [9]. The output of the layer is computed as*

$$y_t = \phi(x_t, u_t) \tag{21}$$

*where*

$$\phi \colon \mathbb{C}^m \times \mathbb{R}^d \to \mathbb{R}^d, \quad (x_t, u_t) \mapsto \mathrm{Mix}_1(\Re(\mathrm{Mix}_2(x_t, u_t)), u_t) \tag{22}$$

$\mathrm{Mix}_1$ *and* $\mathrm{Mix}_2$ *contain linear maps and a non-linearity (either* silu *or* softplus*).*[7] *Note that in our implementation, unlike [9], we do not apply normalization between the two* $\mathrm{Mix}$ *blocks but before the input enters the layer (see Def. 4). For ease of notation, we subsume $C_t$ into* $\mathrm{Mix}_2$ *without loss of generality.* $\mathrm{Mix}_2$ *also usually contains a convolution of the input before the SSM recurrence, which we ignore in expressivity analyses following [11, Remark 18].*

**Definition 2** (Mamba layer)**.** *A Mamba layer is an SSM layer where $A_t$ and $B_t$, are input-dependent and real-valued,*[8] *and, additionally, $A_t \in \mathbb{R}^+$ is non-negative.*

**Definition 3** (AUSSM layer)**.** *An AUSSM layer is an SSM layer where $B_t$ and $C_t$ are fixed constant functions (not input dependent) and $A_t$ is input dependent, complex valued, and each entry has unit magnitude, i.e.,*

$$\forall j \in [d], \quad |A_{t,j}| = \sqrt{\Re(A_{t,j})^2 + \Im(A_{t,j})^2} = 1$$

**Definition 4** (Full SSM)**.** *For a full SSM, we usually stack multiple layers $(1, \ldots, L)$ on top of each other, and indicate the layer we mean by a superscript, e.g., $x_t^{(\ell)}$ is the hidden state at time $t$ in layer $\ell$. The input to the first layer $u_t^{(1)}$ is some embedding of the input of the full SSM computed by some injective embedding function $e : \Sigma \to \mathbb{R}^d$, where $\Sigma$ is the alphabet of possible input values at a single timestep, and the input to layer $\ell \in [L]$ for $\ell > 1$ is the normalized output of the previous layer $\ell - 1$:*

$$u_t^{(\ell)} = \mathrm{Norm}(y_t^{(\ell-1)}) \tag{23}$$

*We use* $\mathrm{RMSNorm}$ *for the* $\mathrm{Norm}$, *defined by*

$$\mathrm{RMSNorm}(x) = \frac{g \circ x}{\sqrt{\frac{1}{n} \sum_{i=1}^n x_i^2}} \tag{24}$$

*where $x \in \mathbb{R}^n$ and $g \in R^d$ is a learned gain parameter. Importantly, like [9], our implementation uses skip connections between consecutive layers, i.e., for*

$$y^{(\ell)} = \phi(x_t, u_t) + y^{(\ell-1)} \tag{25}$$

*The final layer applies another* $\mathrm{RMSNorm}$ *and then a final output function.*

We now introduce some notions from automata theory that are necessary for our expressivity results.

**Definition 5.** *A deterministic finite-state automaton (FSA) $\mathcal{A}$ is a tuple $(\Sigma, Q, \delta)$ where $\Sigma$ is an alphabet (finite, non-empty set), $Q$ is a finite set of states, and $\delta : Q \times \Sigma \to Q$ is a transition function. The transition function can be lifted from symbols to symbol sequences as*

$$\delta : Q \times \Sigma^* \to Q, \quad \delta(q, \varepsilon) = q, \quad \delta(q, \boldsymbol{\sigma}_{\leq t}) = (\delta(q, \boldsymbol{\sigma}_{<t}), \sigma_t)$$

*where $\varepsilon$ is the empty string, $\Sigma^*$ is the Kleene closure over $\Sigma$, and we use boldface to mark sequences of zero or more symbols from $\Sigma^*$.*

The extended transition function $\delta$ forms a transformation monoid under composition, called the *transition monoid* of the FSA.

**Definition 6.** *A set-reset automaton is an FSA whose transition function maps all states to a single state for each input symbol, that is, $\forall \sigma \in \Sigma, \exists p \in Q$ s.t.*

$$\delta(q, \sigma) = p, \quad \forall q \in Q$$

---

[7]Here, $\mathrm{silu}(x) = \frac{x}{1 + \exp(-x)}$ and $\mathrm{softplus}(x) = x \log(1 + \exp(x))$

[8]Note that in Mamba, $B_t$ is directly a function of the input while $A_t$ is input dependent through $\Delta$, which is itself a (non-linear) function of the input.

Note that the transition monoid of a set-reset automaton is aperiodic [67].

**Definition 7.** *A cyclic group automaton is an automaton whose transitions are permutations over states, where every input symbol acts as some power of a fixed $k$-cycle with $k = |Q|$. That is, for every symbol $\sigma \in \Sigma$, the symbol-specific transition map $\delta_\sigma : Q \to Q$ is a bijection, and at least one of the symbols forms a cycle of order exactly $k$, i.e. for some $a \in \Sigma$, $\delta_a^k = \mathrm{id}$ and $\delta_a^n \neq \mathrm{id}\, \forall n \in [1, k-1]$. All other symbol-transition matrices are powers of the same $k$-cycle, i.e., $\forall b \in \Sigma, \delta_b = \delta_a^n$ for some $n \in [0, k-1]$, where $\delta_a^0 = \mathrm{id}$.*

The transition monoid of a $k$-cyclic group automaton is the cyclic group $C_k$ [68].

We start by showing that our AUSSM architecture overcomes the limitation of most SSMs pointed out in [11] by showing that it can perform modulo counting, and therefore, can simulate cyclic group automata.

**Lemma 1.** *For any $k \in \mathbb{Z}^+$, one can construct a single-layer AUSSM that counts modulo $k$, which means AUSSMs can simulate arbitrary cyclic group automata.*

*Proof.* Let $\mathcal{A} = (\Sigma, Q, \delta)$ be a cyclic group automaton. Now we define the input alphabet of the AUSSM to be $\Sigma$ and choose its hidden dimension to be $d = |\Sigma|$. Let $a \in \Sigma$ be the symbol whose transition function $\delta_a$ has order $k$. Then we set the parameters of the AUSSM as follows: Let $B(u) = 0\, \forall u = e(\sigma), \sigma \in \Sigma$. Let $A(e(a)) = \exp(2\pi i/k)$. For each other symbol $b \in \Sigma$, we know there exists $m \in [0, k-1]$ such that $\delta_b = \delta_a^n$, so we can set $A(e(a)) = \exp(2\pi i n/k)$. Now, there is a trivial isomorphism $\psi$ between the values of $x$ and the states of the FSA $\mathcal{A}$: Just define $\psi \colon \{\exp(2\pi i n/k) \mid n \in [k]\} \to \mathbb{Z}/k\mathbb{Z}, \quad \exp(2\pi i n/k) \mapsto n$, which maps every hidden state to the corresponding state of the automaton (arranged in the order of cycle traversal by $\delta_a$). Now there are $n$ distinct possible hidden states which can be read out at logarithmic precision. $\qquad\square$

**A note on numerical precision.** Floating-point operations introduce rounding errors when computing the exponential function and repeated products thereof. A single complex multiplication introduces a relative error of at most $\sqrt{5}u$ [69],[9] where $u$ is the unit roundoff ($u = 2^{-25}$ for 32-bit single and $u = 2^{-53}$ for 64-bit double precision). This yields relative error bounds of $\sqrt{5} \cdot 2^{-24}$ and $\sqrt{5} \cdot 2^{-53}$ respectively. This means after $N$ multiplications, the accumulated relative error is approximately $\sqrt{5}uN$ to the first order. Two adjacent $k$th roots of unity are separated by a $2\pi/k$ segment of the unit circle; by the chord theorem, the distance between them is $\Delta = 2\sin(\pi/k) \approx 2\pi/k$. Approximations start overlapping if the accumulated error surpasses $\Delta/2$, which occurs when:

$$N \geq \frac{\Delta}{2\sqrt{5}u} \approx \frac{\pi}{\sqrt{5}uk} \approx \frac{1.26 \times 10^{16}}{k} \tag{26}$$

For example, with 64-bit double precision and a modulo counter as large as $k = 10^6$, it would take over 12 billion tokens ($N \approx 1.26 \times 10^{10}$) for counts to become indistinguishable. This exceeds the sequence length of most datasets currently used in practice and is an order of magnitude larger than the human genome ($\approx 3 \times 10^9$ base pairs). This means that whenever higher counters or longer sequence lengths are required, one can simply switch to the next higher precision. Since bit-depth is inversely proportional to the logarithm of $u$, we only require logarithmic precision in the sequence length.

The second requirement for transcending the expressivity limits of common SSMs is the ability to implement cascade products of FSAs (see [23, 67, 70] for more details on cascade products):

**Definition 8.** *Let $\mathcal{A}_1 = (\Sigma_1, Q_1, \delta_1), \mathcal{A}_2 = (\Sigma_2, Q_2, \delta_2)$ be FSAs such that $\Sigma_2 = Q_1 \times \Sigma_1$. Then the cascade product $\mathcal{A}_1 \circ \mathcal{A}_2$ is the FSA $\mathcal{C} = (\Sigma_1, Q_1 \times Q_2, \delta_c)$ with $\delta_c$ defined as*

$$\delta_c((q_1, q_2), \sigma) = (\delta_2(q_1, (q_2, \sigma)), \delta_1(q_1, \sigma)) \tag{27}$$

Here, we use tuples of states taken from the state sets of the component FSAs to denote the state of the cascade. Intuitively, the state of the cascade at any given time is the combination of the states that the component FSAs are in at that point.

---

[9] Assuming no underflow, overflow, or subnormal numbers.

Note that for transitioning to the next state, $\mathcal{A}_2$ requires access to the state that $\mathcal{A}_1$ was in before starting the current transition, meaning at time $t$, an FSA higher up in a cascade needs access to the state the lower-level FSAs were in at time $t-1$.

We will hence use the following crucial fact [11] used for constructing FSA cascade products in Mamba SSMs:

**Fact 2** (Sarrof et al. [11], Lemma 17). *For any alphabet $\Sigma$ there exists a single-layer Mamba SSM such that the last-but-one input symbol can be read out from the hidden state at finite precision.*

We will also need the following fact about our hybrid architecture, allowing us to disregard the particular alternating ordering of layer types:

**Note 3.** *We can always add an idempotent Mamba or AUSSM layer in the cascade without changing the model's behavior. This can be done by setting the output projection of the SSM block in question to map everything to zero. Then the input to the next layer will just be the output of the last but one layer (via the skip connection). This means that for any Mamba or AUSSM with a specific behavior, there is a hybrid AUSSM+Mamba with the same behavior.*

Now we have the necessary building blocks to show that our construction fulfills the main requirement for increased expressivity, the ability to implement cascades of the two SSM layer types:

**Lemma 2.** *An SSM consisting of interleaved Mamba and AUSSM blocks (hybrid Mamba+AUSSM) can implement cascade products of automata simulated by Mamba SSMs and AUSSMs.*

*Proof.* We want to show that the hybrid Mamba+AUSSM architecture with alternating Mamba and AUSSM layers can implement cascade products of FSAs. In the following, we take a hybrid Mamba+AUSSM to mean a stack of alternating Mamba and AUSSM layers, ignoring the initial encoding and final normalization and output map. Without loss of generality, assume that the first layer is always an AUSSM layer, and the last layer is always a Mamba layer (we can achieve this by adding idempotent layers where necessary, see Note 3).

Also note that, by Note 3, any Mamba SSM and any AUSSM can be converted to an equivalent hybrid Mamba+AUSSM.

It remains to be shown that a hybrid Mamba+AUSSM can simulate the cascade of two FSAs simulated by hybrid Mamba+AUSSMs. This is simply an extension of [11, Lemma 19] to our hybrid Mamba+AUSSM architecture.

Let $\mathcal{A}_1 = (\Sigma_1, Q_1, \delta_1)$, $\mathcal{A}_2 = (\Sigma_2, Q_2, \delta_2)$ be FSAs such that $\Sigma_2 = Q_1 \times \Sigma_1$. Assume that there are hybrid Mamba+AUSSM models $S_1, S_2$ that map input sequences $(x_1, x_2, \ldots, x_T)$ to the sequences of states under $A_1$, $A_2$, at logarithmic precision.[10]

Let $S_c$ be the hybrid Mamba+AUSSM we want to simulate the cascade $A_1 \circ A_2$. The lower layers of $S_c$ are just the layers of $S_1$. We add $d$ dimensions that just copy the input via a skip connection. We then add a Mamba layer (preceded by an idempotent AUSSM layer) that reads out the second-to-last output of $S_1$ in new dimensions (by Fact 2), while again forwarding the input via the skip connection. Here, we also add a dummy dimension that is always 1, which avoids normalization, making different inputs indistinguishable. Now we have the input and the second-to-last output of $S_1$, corresponding to the last state of $A_1$. Now the remaining layers of $S_c$ are just those of $S_2$, which take this input and compute the transition and state of $A_2$, again adding dimensions such that the state of $A_2$ is separate from the state of $A_1$ and the input to the overall SSM. Now, $S_c$ maps each w to the state sequence under $A_1 \circ A_2$, again at logarithmic precision. This can be inductively extended to a cascade product of arbitrarily many FSAs. □

**Fact 3** (Consequence of Krohn-Rhodes Theorem [23] and the decomposition series of groups [27]). *Any solvable language is recognized by a cascade of set-reset and cyclic group automata.*

**Theorem 2.** *Hybrid Mamba+AUSSM can recognize any solvable language, that is, any regular language whose syntactic monoid does not contain non-solvable subgroups.*

*Proof.* By Lem. 1, an AUSSM can simulate cyclic group automata. By [11, Lem. 19], a Mamba SSM can simulate set-reset automata. By Lem. 2, hybrid AUSSM+Mamba can simulate a cascade

---

[10]Note here we implicitly assume a bijection exists between intervals on $\mathbb{R}^d$ (the input at time $t$, $x_t$) and the alphabet symbols of the relevant FSA.

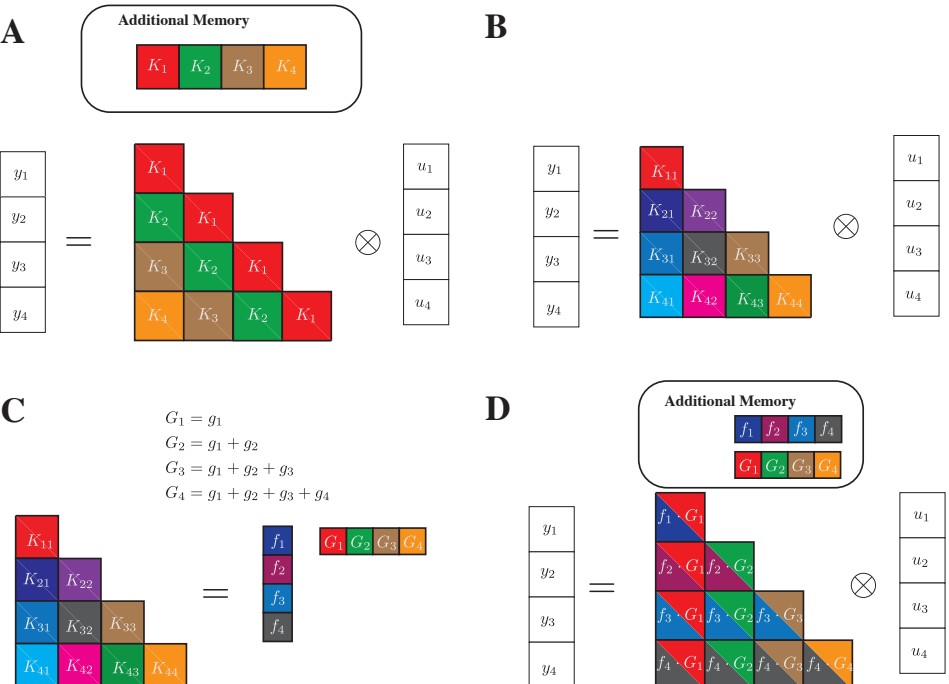

Figure 3: **Space Complexity of SSM formulations**: The figure illustrates an example convolution kernel for an SSM provided with four inputs at different timesteps ($u_t$). The convolution is visualized as a matrix multiplication operation over the input sequence. **A**. In LTI SSMs, the convolution kernel ($K_1, K_2, K_3, K_4$) is precomputed and applied to the input at different timesteps to obtain the output. **B**. In general LTV SSMs with time-varying recurrence, the convolution kernel has $O(L^2)$ elements, each unique to the input and output being considered at each timestep. The use of convolution in this scenario leads to quadratic complexity in space (akin to the transformers). **C**. In the separable convolution case, the quadratic matrix of the general SSM can actually be obtained by the outer product between $f_t$ for each timestep and the cumulative sums of a function $g_k$ independent of $t$. **D**. Computing the convolution kernel can be achieved in just an additional $O(2L)$ space.

of automata simulated by Mamba and AUSSM SSMs. Together with Fact 3, this means that hybrid AUSSM+Mamba can recognize any solvable language. $\qquad\square$

**The importance of counting for other tasks.** Note that the ability to count modulo $k$ does not just allow SSMs to model regular languages but also to approximate languages higher up on the Chomsky hierarchy. For example, it allows the recognition or generation of bounded Dyck languages, i.e., the correct parenthesization up to a certain depth (see [71] in the case of RNNs). Even context-sensitive language tasks can benefit from counting: For instance, sorting a sequence (the bucket sort task in §5) can be done by maintaining counters for all alphabet symbols and then outputting the symbols in order, according to their count (see counting sort and direct-address tables [72, Chapters 8 and 11]). Note that this works as long as the number of occurrences of any given symbol is smaller than the highest count expressible by the SSM, e.g., $k$ when using modulo $k$ counting.

## F Complexity Analysis

SSMs leverage logarithmic complexity algorithms like FFT and parallel prefix sum to compute the convolution. Prior to this, the convolution kernel needs to be pre-computed and stored, which is the main bottleneck in computing the convolution. We will show below the space complexities for computing and storing the convolutions. Further, we show how the quadratic space complexity blowup of pure LTV systems can be managed using the separable convolution framework.

### F.1 SSM Convolution

The convolution operation of a general SSM is given by the following

$$y(t) = \sum_{k \leq t} C_t \left( A_{t-1}...A_{k+2}A_{k+1} \right) B_k \, u(k) \tag{28}$$

There are two cases for the above convolution we consider:

**Linear Time Invariant (LTI)** : In the LTI case, the matrices in the SSM are constant over time, and the following holds

$$y(t) = \sum_{k \leq t} CA^{t-1-k}B \, u(k) \tag{29}$$

Now, the convolution kernel $K(t,k) = C \, A^{t-1-k} B$ can be precomputed, and since $A^{t-k-1}$ is common for many settings of $t$ and $k$ for which their difference is constant, the weights can be shared. In fact, there are only $O(L)$ unique entries in the convolution kernel (see Figure 3 A). The other entries are duplicates of these entries. Once the convolution kernel is obtained, efficient algorithms like FFT or Parallel Scan can be used to compute the convolution in $O(\log L)$ time for each dimension, for a total of $O(L \log(L))$ time complexity. Therefore, the total time complexity for computing the kernel is $O(L \log L)$ with a space complexity of $O(L)$.

**Linear Time Varying (LTV)** : In the LTV case, the matrices in the SSM can vary over time. This introduces additional complexity in representing the convolution kernel in $O(L^2)$ space, matching the quadratic complexity of computing self-attention in transformers. The reason for the quadratic complexity is that the entries in the convolution kernel $K(t,k)$ are unique for each setting of $t, k$. In the case of separable convolution kernels (e.g, the case of simultaneously diagonalizable matrices), the resulting $K(t,k)$ matrix has a further rank-1 factorization (this is discussed in detail in the main text). This factorization enables the convolution kernel to be represented with only an additional $O(2L)$ memory, where the 2 factor comes from each vector element in the outer product.

### F.2 Parallel Scan

The reason for precomputing the convolution kernel is that we can apply one of the fast convolution algorithms - FFT or parallel scan. In our case, we perform the parallel prefix sums for computing cumulative sums. Here, we analyze the time and space complexity of the parallel prefix sum (scan) algorithm, where the goal is to compute the prefix sums of an array $A = [a_0, a_1, \ldots, a_{L-1}]$ such that the output array $S$ satisfies

$$S_i = \sum_{j=0}^{i} a_j \quad \text{for } 0 \leq i, j < L. \tag{30}$$

We assume a parallel computation model such as the PRAM (Parallel Random Access Machine) or a shared-memory model, and we are given $P$ processors.

The parallel prefix sum algorithm typically consists of two main phases:

1. **Upsweep phase (Reduction):** Build a binary tree over the array and compute partial sums from leaves to the root.
2. **Downsweep phase:** Propagate prefix sums from the root back down the tree to compute the final result.

Both phases traverse a binary tree structure of height $\log_2 L$, assuming for simplicity that $L$ is a power of two. Each level of the tree can be processed in parallel.

**Work.** The total number of operations (work) in both phases is:

$$W(L) = \underbrace{(L-1)}_{\text{upsweep}} + \underbrace{(L-1)}_{\text{downsweep}} = 2L - 2 = \mathcal{O}(L). \tag{31}$$

This is the same amount of work as the sequential prefix sum algorithm, which confirms that the parallel algorithm is work-efficient.

**Time Complexity with $P$ Processors.** Using Brent's Theorem (work-span model), the parallel time $T_P$ on $P$ processors is bounded by:

$$T_P \leq \frac{W(L)}{P} + S(L) = \mathcal{O}\left(\frac{L}{P} + \log L\right). \tag{32}$$

This means that when the number of parallel processors grows in the sequence length according to $P = \Theta(L/\log L)$, the parallel prefix sum runs in optimal time $\mathcal{O}(\log L)$.

**Space Complexity** The space used by the algorithm includes:

- The original input array $A$, of size $L$.
- An auxiliary array to store intermediate results, typically of size $L$.
- Additional temporary variables per processor (constant per processor).

Hence, the total space complexity is:

$$\mathcal{O}(L + P) = \mathcal{O}(L) \quad \text{(since typically } P \leq L\text{)}. \tag{33}$$

It is important to note that although the algorithm requires additional $O(L)$ space for the auxiliary variables, the CUDA kernel implementation hides these variables within the multiprocessor registers and shared memory. As a result, this complexity does not show up in the plots of either Mamba or auSSM. Existing GPU hardware for the 2080ti enables parallel processing of sequences up to $L = 2048$. For longer sequences, the input is chunked into batches of $L = 2048$.

## G Implementation

The theoretical analysis of the separable kernel formulation shows that the adaptive kernel can be implemented in only an additional linear space. However, the factor associated with the linear space is $bdn$, where $b$ is the batch size, $d$ is the input dimension, and $n$ is the hidden state dimension. In this section, we first show a PyTorch implementation of the AUSSM kernel and Mamba kernel that can be easily coded, with the higher cost of the constant factors. Next, we show how we implement the AUSSM kernel in practice so that the additional complexity is hidden within the computations of a CUDA kernel.

### G.1 PyTorch

One of the most useful aspects of the theory of separable convolutions is that there is a relatively efficient PyTorch formulation for computing SSM kernels, even when the SSM is partially/fully time varying. However, an additional constant-time penalty will be incurred. Nevertheless, the existence of such an implementation will still be interesting as it can enable fast prototyping of LTV SSMs, without dealing with the complexity of building a CUDA kernel. Here, we show two PyTorch implementations of the partial LTV Mamba kernel and the separable AUSSM kernel.

```python
def mamba_ssm(u, dt, A, B, C, D, z):
    """
    params:
        u: input Tensor (b,d,l)
        dt: Delta Tensor (b,d,l)
        A: Tensor (n)
        B: Tensor (b,n,l)
        C: Tensor (b,n,l)
        D: Tensor (d)
        z: Tensor (b,d,l)
    Returns:
        y: (b, d, l)
    """
    A = einsum(A, dt, "n,bdl->bdnl")
    G = torch.cumsum(axis=1)

    g = einsum(exp(-G), dt, B, u, "bdnl,bdl,bnl,bdl->bdnl"
    g = torch.cumsum(g, axis=-1)
    f = einsum(C, exp(G), "bnl,bdnl->bdnl")

    y = einsum(f, g, "bdnl,bdnl->bdl") + D * u
    y = y * F.silu(z)

    return y
```

The implementation of Mamba using the separable kernel formulation has fewer than 10 lines of PyTorch code. The PyTorch implementation of AUSSM is similar, except now we have to account for the time-varying $A$ matrix, and $B$ and $C$ are relaxed.

```python
def aussm(u, dt, chi, B, C, D, z):
    """
    params:
        u: input Tensor (b,d,l)
        dt: Delta Tensor (b,d,l)
        chi: adaptivity matrix (d,n,d)
        B: Tensor (n)
        C: Tensor (n)
        D: Tensor (d)
        z: Tensor (b,d,l)
    Returns:
        y: (b,d,l)
    """
    A = einsum(chi, u, "dnr,blr->bldn")
    A = einsum(dt, A, "bdl,bldn-bldn")
    G = torch.cumsum(axis=1)

    g = einsum(exp(-G), dt, B, u, "bdnl,bdl,n,bdl->bdnl"
    g = torch.cumsum(g, axis=-1)
    f = einsum(C, exp(G), "n,bdnl->bdnl")

    y = einsum(f, g, "bdnl,bdnl->bdl") + D * u
    y = y * F.silu(z)

    return y
```

In this implementation, Mamba and PyTorch have the same space and time complexity as the hidden state is realized for both, albeit at only a fraction of the cost.

### G.2 CUDA Kernel

In pure PyTorch, the additional complexity of realizing the hidden state is unavoidable, even though the computation does not have quadratic memory costs. The additional complexity of realizing the hidden state can be avoided by creating a CUDA kernel for the AUSSM equation. We use the following equation for the AUSSM, which we introduced in the main text:

$$y_{ti} = \Re \left[ \sum_{k \leq t} \sum_j C_j \exp\left( \mathbf{i} \sum_{l \leq t} \theta_{A_{lij}} \right) \frac{\Delta_{ki} B_j}{\exp\left( \mathbf{i} \sum_{l \leq k} \theta_{A_{lij}} \right)} u_i(k) \right]. \tag{34}$$

Each thread of the CUDA implementation computes the array inside the nested summation, which results in $O(L)$ memory requirement for storing each of the variables $(A, f, g)$ for the forward pass. These variables are not realized at the same time in the GPU memory, but in registers within the streaming multiprocessors (SM), each processor holding 4 to 16 items of each array. For the 2080Ti GPU, we ran the CUDA kernel on, the allowable maximum sequence length that can be processed by the kernel was 2048, after which the register and shared memory costs start to show up. We found that this sequence length is ideal for the hardware and tasks we tested on. The separable convolution trick is not restricted by the hardware and can scale well for GPUs that can be released in the future with larger registers and shared memory resources.

**Backpropagation:** For the CUDA kernel, we implemented a custom backpropagation operation. Implementing backpropagation requires the variables computed during the storage to be stored, which creates issues because the reason we are writing the CUDA kernel is so that we do not have to realize the memory-intensive hidden state. We therefore recompute the forward pass during backpropagation. The low complexity of implementing the AUSSM in CUDA means the recomputation does not incur a heavy penalty.

Table 6: **Best Hyperparameters.**

| Task Group | Task | Layers | d | n | weight decay | learning rate |
|---|---|---|---|---|---|---|
| Algorithmic | repetition | ma | 64 | 32 | 0.0 | 0.01 |
| | bucket sort | am | 64 | 32 | 0.0 | 0.01 |
| | majority count | ma | 64 | 32 | 0.1 | 0.01 |
| | majority | ma | 64 | 32 | 0.1 | 0.01 |
| | solve equation | ma | 64 | 32 | 0.0 | 0.01 |
| | mod arith | am | 16 | 8 | 0.0 | 0.01 |
| | mod arith wo bra | ma | 8 | 16 | 0.0 | 0.01 |
| | cycle nav | ma | 16 | 8 | 0.0 | 0.01 |
| | parity | ma | 16 | 8 | 0.0 | 0.01 |
| Timeseries Classification | Heartbeat | ma | 64 | 64 | 0.0 | 0.0001 |
| | SCP1 | amma | 16 | 128 | 0.0 | 0.001 |
| | SCP2 | ma | 16 | 128 | 0.0 | 0.0001 |
| | Ethanol | ammama | 16 | 64 | 0.001 | 0.00001 |
| | Motor | ma | 16 | 128 | 0.0 | 0.0001 |
| | Worms | amma | 16 | 16 | 0.0 | 0.001 |
| Timeseries Regression | weather | ma | 16 | 128 | 0.0 | 0.001 |

## H  Experiments

We conduct three sets of experiments: (1) to evaluate the time/memory complexities of the different AUSSM implementations, (2) to evaluate the performance of AUSSM in algorithmic tasks enabling insights into the expressive power, and (3) to evaluate real-world performance implications in a range of long time series benchmarks. For each of the tasks involving training models (2 and 3), we perform two pipeline processes to obtain the final test accuracies. The first pipeline is the training and model selection pipeline with only the training and validation sets that are preselected based on the same criteria used by prior literature. The second pipeline is the test pipeline and is entirely separate and performed starting 10 days prior to paper submission to avoid model selection based on the test results. The classification tasks are evaluated using the scaled test accuracy metric, where the obtained accuracy values are scaled with respect to the baseline performance of a uniform random distribution, as shown below.

$$\text{scaled accuracy score} = \frac{\text{test accuracy score} - \text{baseline accuracy score}}{1 - \text{baseline accuracy score}}$$

All the models were run in a supercomputing cluster, where we used 40 2080Ti GPUs for all except the dataset `Eigenworms` dataset that required higher memory. This is the lowest GPU available in the cluster, with at least a CUDA compute of 7.5 required to run the Mamba and AUSSM CUDA kernels. For a larger memory `Eigenworms` workload, we used the L4 GPU, which has a VRAM of 23GB. Higher VRAM GPUs were available in the cluster, but they were in high demand and unnecessary, as our optimized CUDA kernel was able to handle even the large-scale tasks in modest hardware.

### H.1  Scalability Evaluation

To evaluate scalability in a fair manner, we report only the time spent in computations, ignoring the latencies associated with moving variables between the GPU and the CPU. This provides a fair evaluation of the algorithmic performance. 5 runs are used to warm up the GPU before starting the evaluation to remove transient start-up effects. The run-time values are averaged over 50 runs, where each run computes a forward and backward pass for each of the implementations. The peak memory used during each run is also similarly recorded and averaged for each of the 50 runs.

### H.2  Time Series benchmark

For time series classification and regression benchmarks, we follow the train-validation protocol for model selection, following prior works on the benchmark. For testing, we modified the procedure

as the five arbitrary random seeds used to evaluate test performance in prior works may introduce unwanted biases due to the low number of random samples. Also, prior works used JAX for implementations, while we used PyTorch, and the random seed does not create the same train-validation-test sets due to differences in the pseudorandom number generators. We thus decided to evaluate on train-validation-test splits created with 20 different seeds. We anticipated that the higher samples would help in providing a better estimation of the test accuracy than what the five arbitrary seeds provide. For each task, we performed a hyperparameter search over the following grid: $d \in \{16, 64, 128\}, n \in \{16, 64, 128\}$, learning rate $\in \{0.00001, 0.0001, 0.001\}$, and five different seeds for model selection. The model hyperparameters with the highest mean validation accuracy are chosen for evaluation in the test set.

## H.3 Algorithmic Tasks

For algorithmic tasks, we used the results from [31] for comparing against baseline models. We used a grid search for hyperparameter tuning with a grid search over $d \in \{8, 16, 32, 64\}, n \in \{8, 16, 32\}$, weight decay $\in \{0.0, 0.001, 0.01\}$, learning rate in $\{0.0001, 0.001, 0.01\}$ and five seeds. The batch size was fixed at 256. For pure AUSSM blocks, we tested networks with a depth of 2, 4, and 6. For hybrid AUSSM blocks, we tested all possible 2-block configurations of Mamba (represented as m) and AUSSM blocks (represented as a) - {ma, am, mm, aa}. For each of the evaluated algorithmic tasks, we randomly sampled 10000 samples from a train set up to length-40 sequences. The validation set is sampled independently from 40-256 sequence lengths and had 1,000 samples. The test set had 10,000 samples from sequences of up to 256 sequence lengths.

The tasks use the same vocabulary size and configuration used in [31]. Some samples from the tasks are shown below as a timeline. Here, the mask is applied to the output to determine the output of interest for computing the loss and output.

Task: repetition

| time | 0 | 1 | 2 | 3 | 4 | 5 | 6 | 7 | 8 | 9 | 10 |
|---|---|---|---|---|---|---|---|---|---|---|---|
| input | 3 | 5 | 0 | 7 | 3 | ACT | 3 | 5 | 0 | 7 | 3 |
| output | 5 | 0 | 7 | 3 | ACT | 3 | 5 | 0 | 7 | 3 | PAD |
| mask | 0 | 0 | 0 | 0 | 0 | 1 | 1 | 1 | 1 | 1 | 0 |

Task: bucketsort

| time | 0 | 1 | 2 | 3 | 4 | 5 | 6 | 7 | 8 | 9 | 10 |
|---|---|---|---|---|---|---|---|---|---|---|---|
| input | 3 | 5 | 0 | 7 | 3 | ACT | 0 | 3 | 3 | 5 | 7 |
| output | 5 | 0 | 7 | 3 | ACT | 0 | 3 | 3 | 5 | 7 | PAD |
| mask | 0 | 0 | 0 | 0 | 0 | 1 | 1 | 1 | 1 | 1 | 0 |

Task: modarithmeticwobraces

| time | 0 | 1 | 2 | 3 | 4 | 5 | 6 | 7 | 8 | 9 | 10 |
|---|---|---|---|---|---|---|---|---|---|---|---|
| input | 0 | * | 2 | − | 6 | − | 7 | − | 0 | = | 5 |
| output | * | 2 | − | 6 | − | 7 | − | 0 | = | 5 | PAD |
| mask | 0 | 0 | 0 | 0 | 0 | 0 | 0 | 0 | 0 | 1 | 0 |

Task: cyclenav

| time | 0 | 1 | 2 | 3 | 4 | 5 | 6 | 7 | 8 | 9 | 10 |
|---|---|---|---|---|---|---|---|---|---|---|---|
| input | +1 | STAY | +1 | −1 | +1 | −1 | −1 | −1 | +1 | 0 | PAD |
| output | STAY | +1 | −1 | +1 | −1 | −1 | −1 | +1 | 0 | PAD | PAD |
| mask | 0 | 0 | 0 | 0 | 0 | 0 | 0 | 0 | 1 | 0 | 0 |

Task: modarithmetic

| time | 0 | 1 | 2 | 3 | 4 | 5 | 6 | 7 | 8 | 9 | 10 |
|---|---|---|---|---|---|---|---|---|---|---|---|
| input | ( | ( | 3 | − | 3 | ) | − | 4 | ) | = | 3 |
| output | ( | 3 | − | 3 | ) | − | 4 | ) | = | 3 | PAD |
| mask | 0 | 0 | 0 | 0 | 0 | 0 | 0 | 0 | 0 | 1 | 0 |

Task: solveequation

| time | 0 | 1 | 2 | 3 | 4 | 5 | 6 | 7 | 8 | 9 | 10 |
|---|---|---|---|---|---|---|---|---|---|---|---|
| input | x | = | ( | 2 | + | 1 | ) | ACT | 3 | PAD | PAD |
| output | = | ( | 2 | + | 1 | ) | ACT | 3 | PAD | PAD | PAD |
| mask | 0 | 0 | 0 | 0 | 0 | 0 | 0 | 1 | 0 | 0 | 0 |

Task: parity

| time | 0 | 1 | 2 | 3 | 4 | 5 | 6 | 7 | 8 | 9 | 10 |
|---|---|---|---|---|---|---|---|---|---|---|---|
| input | 1 | 1 | 0 | 1 | 1 | 1 | 1 | 1 | 1 | 1 | 0 |
| output | 1 | 0 | 1 | 1 | 1 | 1 | 1 | 1 | 1 | 0 | 0 |
| mask | 0 | 0 | 0 | 0 | 0 | 0 | 0 | 0 | 0 | 0 | 1 |

Task: majoritycount

| time | 0 | 1 | 2 | 3 | 4 | 5 | 6 | 7 | 8 | 9 | 10 |
|---|---|---|---|---|---|---|---|---|---|---|---|
| input | 45 | 56 | 51 | 43 | 51 | 34 | 10 | 46 | 54 | 44 | 56 |
| output | 56 | 51 | 43 | 51 | 34 | 10 | 46 | 54 | 44 | 56 | 2 |
| mask | 0 | 0 | 0 | 0 | 0 | 0 | 0 | 0 | 0 | 0 | 1 |

Task: majority

| time | 0 | 1 | 2 | 3 | 4 | 5 | 6 | 7 | 8 | 9 | 10 |
|---|---|---|---|---|---|---|---|---|---|---|---|
| input | 45 | 56 | 51 | 43 | 51 | 34 | 10 | 46 | 54 | 44 | 56 |
| output | 56 | 51 | 43 | 51 | 34 | 10 | 46 | 54 | 44 | 56 | 51 |
| mask | 0 | 0 | 0 | 0 | 0 | 0 | 0 | 0 | 0 | 0 | 1 |

Task: set

| time | 0 | 1 | 2 | 3 | 4 | 5 | 6 | 7 | 8 | 9 | 10 |
|---|---|---|---|---|---|---|---|---|---|---|---|
| input | 3 | 5 | 0 | 7 | 3 | ACT | 0 | 3 | 5 | 7 | PAD |
| output | 5 | 0 | 7 | 3 | ACT | 0 | 3 | 5 | 7 | PAD | PAD |
| mask | 0 | 0 | 0 | 0 | 0 | 1 | 1 | 1 | 1 | 0 | 0 |

