# OpenReview forum: "Bridging Expressivity and Scalability with Adaptive Unitary SSMs"
_NeurIPS.cc/2025/Conference — NeurIPS 2025 poster_

### Official Review · Reviewer_MtYg · 2025-06-21

**Clarity:** 2
**Significance:** 2
**Originality:** 2
**Rating:** 4
**Confidence:** 4

**Summary:**

The presented model extends the eigenvalues of the state-transition matrices of Mamba 1 from the positive reals to the unit circle. The authors demonstrate that this change allows the model to learn state-tracking tasks such as parity or cyclic navigation which Mamba 1 using only real eigenvalues provable cannot learn in a length extrapolatible way.

**Questions:**

-

**Ethical Concerns:**

["NO or VERY MINOR ethics concerns only"]

**Final Justification:**

The authors addressed my questions. However, I think the paper is somewhat misleading in that it makes the case for using parallel-scans while these have been shown to be much slower on current hardware than chunk-wise parallel forms.

**Limitations:**

Yes.

**Paper Formatting Concerns:**

-

**Quality:**

2

**Strengths And Weaknesses:**

## Strengths

- Extending the eigenvalues of the diagonal state-transition matrix to complex values makes sense to me, however I am unsure about keeping them purely unitary, see weaknesses.
- The experiments demonstrate that this allows them to model state-tracking tasks which are unsolvable by Mamba 1 with purely real and positive eigenvalues.
- The architecture shows promise for long-term time-series forecasting.


## Weaknesses
- Models such as RWKV-7 and DeltaNet use non-diagonal state-transition matrices which were shown to be significantly better at in-context retrieval and also state-tracking when using negative eigenvalues as demonstrated by Grazzi et al. 2025. Since these are also linear RNNs it would be important to compare to them. In particular DeltaNet with negative eigenvalues can solve modular arithmetic w/o brackets (Grazzi et al. 2025) in addition to parity. Grazzi et al. 2025 also provide code to run Mamba with negative eigenvalues which you could compare to in order to show the need for unitary eigenvalues.

-  It is not clear to me why the eigenvalues need to be unitary and could not also lie in the interior of the unit circle in order to allow the model to maintain signal when using the boundaries of the circle or attenuate signal when using more of the interior of the circle in order to forget better.

- As I understand the critical difference between Mamba and the proposed AUSSM model is that the eigenvalue range is extended from the positive reals in Mamba to the complex unit circle. Since Mamba 1 already used complex eigenvalues in the state-transition matrix for some of their experiments on audio sequences, please compare with their implementation to isolate your contribution.

- I think the key changes to mamba could be emphasized more straightforwardly. The usefulness of coining the term "adaptive" is unclear to me, simply calling it a linear RNN with token-dependent diagonal state-transition matrix having unitary entries would be more clear or did I misunderstand?

- Nonlinear Unitary RNNs have a rich research history, please at least give a small literature overview for example on those using rotation or reflection matrices. The following come to mind: Z. Mhammedi, A. Hellicar, A. Rahman, and J. Bailey. Efficient orthogonal parametrisation of recurrent neural networks using householder reflections. Victor D. Dorobantu, Per Andre Stromhaug, and Jess Renteria. DizzyRNN: Reparameterizing Recurrent Neural Networks for Norm-Preserving Backpropagation. L. Jing, Y. Shen, T. Dubcek, J. Peurifoy, S. Skirlo, Y. LeCun, M. Tegmark, and M. Soljacic. Tunable Efficient Unitary Neural Networks (EUNN) and their application to RNNs. Rumen Dangovski, Li Jing, Preslav Nakov, Miśo Tatalović, and Marin Soljačić. Rotational unit of memory: a novel representation unit for rnns with scalable applications. C. Jose, M. Cisse, and F. Fleuret. Kronecker recurrent units.

---

> ### Author Rebuttal · Authors · 2025-07-31
>
> We thank the reviewer for the constructive comments and critical evaluation of our manuscript. Thank you for highlighting AUSSM’s **promise for long term time-series forecasting**. We want to clarify that we make **two extensions** to Mamba - (1) allowing for unitary eigenvalues **and** (2) enabling the eigenvalues to adapt based on the instantaneous inputs. We want to emphasize that the S6 SSM ODE used in Mamba has only B and C adaptive, while A is adaptive only in the discrete form indirectly through a time step factor $\Delta$. Thus, Mamba dynamics can speed up or slow down depending on input, but not fundamentally change. Our AUSSM formulation enables the full $A$ also to be adaptive, enabling input-dependent (adaptive) dynamical behaviors, while encountering minimal penalty (only a small constant factor) to computational efficiency.
>
> We address the concerns of comparing with modified Mamba versions (negative and complex) by adding new formal language result comparisons. We address the concern of a literature review of non-linear unitary RNNs by adding a Review section in our appendix. We summarize the changes below and provide detailed clarifications to each question in the subsequent section.
> ## Relevant Changes
>
> ### 1. Experiments with Negative and Complex versions of Mamba
>
> Thank you for the suggestion to use the general eigenvalue Mamba versions. We implemented new experiments with both Mamba\[-1,1\] from the Grazzi paper and Mamba Complex from the original Mamba paper. Please note that the original Mamba paper did not detail how B and C are defined for the complex version, nor provide code to replicate Mamba results with complex eigenvalue spectrum, as evidenced by this issue#106 in the Mamba github repo. Nevertheless, we implemented our own version by modifying A to have S4D-Lin initialization and increasing the dimensionality of B to account for the imaginary part. We report the results of running the models on the Formal Language Benchmark below
>
> |                   | Mamba with Complex Eigenvalues | Mamba in \[-1,1\] range |
> | ----------------- | ------------------------------ | ----------------------- |
> | repetition             | 0.09                           | 0.1                     |
> | bucket sort          | 0.21                           | 0.91                    |
> | majority count         | 0.19                           | 0.31                    |
> | majority          | 0.13                           | 0.64                    |
> | solve equation             | 0.43                           | 0.24                    |
> | modarithmetic          | 0.12                           | 0.116                   |
> | modarithmetic wo bracket | 0.23                           | 0.24                    |
> | cyclenav          | 0.42                           | 0.91                    |
> | parity            | 0.27                           | 1.0                     |
>
> As reported in prior works, Mamba[-1,1] performs better compared to Mamba on the parity, cyclenav, solve equation, and bucketsort. However, AUSSM and the AUSSM Hybrid model still perform better in most tasks except bucketsort and solve equation. The tested complex version of Mamba performed poorly compared to the Grazzi version, which iterates the finding in the Mamba paper that real-eigenvalue parameterizations performed better. In AUSSM, the additionally introduced adaptivity and unitarity provide sufficient requirements for enabling learnability and expressivity.
>
> ### 2. Literature Review for Non-linear Unitary RNNs
>
> We thank the reviewer for this suggestion. As our contribution was mainly in the development of SSMs, our current version of the paper only contained relevant SSM implementations. We add a new section in the appendix where we provide a detailed review of Non-linear Unitary RNNs. We show a brief snippet here due to character limits in the rebuttal responses.
>
> "Unitary and orthogonal RNNs were introduced to mitigate vanishing and exploding gradients by preserving the norm of hidden states over time. Early work by Arjovsky et al. (2016) proposed Unitary RNNs with parameterizations based on structured unitary matrices, followed by Full-Capacity Unitary RNNs (Wisdom et al., 2016) which optimized directly over the class of all unitary matrices (Jing et al., 2017). Orthogonal RNNs have also been explored using soft constraints (Vorontsov et al., 2017), matrix exponentials of skew-symmetric matrices (Lezcano-Casado & Martínez-Rubio, 2019), and other structured parametrizations such as Householder reflections (Mhammedi et al., 2017)...."
>
> Arjovsky et al. (2016). Unitary Evolution RNN. ICML.
>
> Wisdom et al. (2016). Full-Capacity Unitary RNNs. NeurIPS.
>
> Vorontsov et al. (2017). On orthogonality and learning recurrent networks with long term dependencies. ICML.
>
> Lezcano-Casado & Martínez-Rubio (2019). Cheap Orthogonal Constraints in Neural Networks: A Simple Parametrization of the Orthogonal and Unitary Group. ICML.
>
> Mhammedi et al. (2017). Efficient Orthogonal Parametrization Using Householder Reflections. ICML.
>
> Jing et al. (2017). Tunable Efficient Unitary Neural Networks (EUNN) and their application to RNNs . JMLR.
>
>
>
>
> ## Addressing Weaknesses and Questions
>
> ### Models such as RWKV-7 and DeltaNet use non-diagonal state-transition matrices...
>
> We thank the reviewer for the constructive suggestions to improve the manuscript. To address the suggestions, we have thus added new comparisons to the Grazzi Mamba implementation and Mamba Complex implementation (see #1 in the Changes section). RWKV-7 \[1\] and DeltaNet \[2,3\] both use non-diagonal transition matrices, trading added computation cost for expressivity.  In DeltaNet, a single Householder reflection per step is insufficient for full regular language expressivity or even to perform arbitrary rotations. DeltaProduct \[4\] addresses this by using products of multiple reflections, which can realize arbitrary orthogonal transformations. By contrast, AUSSM can perform rotations cheaply with a single diagonal update. The added computational cost of using non-diagonal matrices is a recurring bottleneck typical of non-linear RNNs. Our focus was to make diagonal SSMs, which have efficient parallel implementations, as expressive as possible while retaining their computational efficiency advantage. From a formal expressivity standpoint, negative eigenvalues in diagonal SSMs are not enough to perform even simple tasks such as mod 3 addition (Grazzi, Thm. 2). Unitary eigenvalues, however, lift the expressivity to all solvable languages, the upper bound possible for diagonal SSMs (assuming TC0 != NC1). Further increase in formal expressivity is only possible using non-diagonal and, hence, a computationally inefficient approach, as we discuss in our Limitations section.
>
> ### It is not clear to me why the eigenvalues need to be unitary and could not also lie in the interior of the unit...
>
> Using unitary eigenvalues in the AUSSM part reduces the number of parameters because only the polar angle is parameterized, improving training efficiency, while adding enough expressivity to allow recognition of all solvable languages. The Mamba part of the hybrid architecture takes care of the memory/forgetting part since it does not have fixed eigenvalues. This is also borne out by our experiments, which show that the hybrid model performs much better than Mamba with complex eigenvalues (which has the same expressive power).
>
> ### As I understand the critical difference between Mamba and the proposed AUSSM...
>
> We make **two changes** to the S6 ODE used in Mamba (real version) - (1) extend the eigenvalue range of A to unitary and (2) enable adaptivity in the eigenvalue spectrum. We thank the reviewer for the suggestion to compare with Mamba1 with complex eigenvalues. We have now added the results from running Mamba1 with complex eigenvalues in the formal language benchmarks (see #1 in the Changes section). Grazzi's version of Mamba performs better compared to the complex Mamba, despite the complex one having better expressivity. This reinforces the result from the Mamba paper that real-valued mamba versions performed better than the complex-valued version. Existing works like Mamba with complex eigenvalues do not have full adaptivity - the recurrence is mildly adaptive in the sense that the eigenvalues can scale up/down depending on inputs, but cannot change sign or become complex. e.g., if we want to rotate left vs right depending on input.
>
> ### The usefulness of coining the term "adaptive" is unclear to me,...
>
> AUSSM is adaptive in the sense that the recurrence structure changes (adapts) to every input presented to it. This goes beyond the selectivity introduced in the S6 SSM used in Mamba. In AUSSM,  the $\Delta$ is not the only driver of time-varying behavior but the recurrent matrix $A$ itself. This way, AUSSM forms a middle ground between the partial LTV structure of Mamba and a full LTV dynamical system and enables the dynamics to adapt (not just scale as in partial LTV Mamba) to changes in the input.
>
> ### Nonlinear Unitary RNNs have a rich research history,...
>
> We thank the reviewer for the suggestion. We have added a new section in the appendix that aims to cover some of the main themes in the design of non-linear unitary RNNs (See Changes #2 above)
>
> ## References
>
> \[1\] Peng et al., 2025. RWKV-7 "Goose" with Expressive Dynamic State Evolution
> \[2\] Schlag et al., 2021. Linear transformers are secretly fast weight programmers. ICML
> \[3\] Grazzi et al., 2025. Unlocking State-Tracking in Linear RNNs Through Negative Eigenvalues. ICLR
> \[4\] Siems et al., 2025. DeltaProduct: Improving State-Tracking in Linear RNNs via Householder Products

---

> > ### Comment · Reviewer_MtYg · 2025-08-03
> >
> > I thank the authors for the detailed rebuttal. I have only two remaining questions that I would appreciate clarification on.
> >
> > ### 1. Experiments with Negative and Complex versions of Mamba
> > Thank you for this very interesting comparison. Could you help me understand why the complex parameterization of Mamba is performing so poorly here? I'm curious about whether the complex parameterization of Mamba subsumes yours, or what the relation is in terms of eigenvalues of its state-transition matrix to yours?
> >
> > ### Models such as RWKV-7 and DeltaNet use non-diagonal state-transition matrices...
> > I agree with you that non-diagonal linear RNNs have expressivity advantages as demonstrated by RWKV-7 and DeltaProduct. However, I would like to respectfully discuss the following statement:
> > > 'Further increase in formal expressivity is only possible using non-diagonal and, hence, a computationally inefficient approach, as we discuss in our Limitations section.'
> >
> > I believe this claim could benefit from some nuancing. See Figure 3 in Yang et al. (2025) for a throughput comparison between e.g. Mamba 1, Mamba 2 and DeltaNet / GatedDeltaNet. Here Mamba 1 is slow due to using a selective-scan (which can't leverage tensor cores and is limited to slower cuda cores). Also note that Mamba 2 and DeltaNet have similar throughput here. The numbers in Figure 3 are outdated and Gated DeltaNet is even faster now in flash-linear attention repository through the UT transform optimizations proposed by Hu et al. (2025).
> >
> > From your source code I understand that you're also leveraging the selective-scan. Would it be possible to provide throughput numbers compared to (Gated) DeltaNet?
> >
> > ### References:
> > S. Yang, J. Kautz, and A. Hatamizadeh. Gated delta networks: Improving mamba2 with delta rule. In The Thirteenth International Conference on Learning Representations (ICLR'25). ICLR, 2025.
> >
> > Hu, Jiaxi et al. "Comba: Improving Bilinear RNNs with Closed-loop Control." ArXiv abs/2506.02475 (2025): n. pag.

---

> > > ### Author Response · Authors · 2025-08-05
> > >
> > > We thank the reviewer for engaging with our rebuttal responses and for the references to contemporaneous work. We will add the discussion to these works in the updated manuscript.
> > >
> > > ## Could you help me understand why the complex parameterization of Mamba is performing so poorly here?
> > >
> > > There is no rigorous proof for why this contradiction occurs, even though Mamba-complex includes Mamba real as a special case with eigenvalue imaginary part exactly 0. We conjecture that there is an interplay with the *learnability* of the Mamba-complex model compared to Mamba real. i.e the expressivity increase in Mamba complex may make it less learnable - there are no free lunches. This experimental result is consistent with earlier work with Mamba that shows Mamba complex performs poorly in comparison to Mamba with real eigenvalue spectrum in language tasks, which may be one of the reasons why the real-valued versions of Mamba are more popular in practical scenarios.
> > >
> > > ## I'm curious about whether the complex parameterization of Mamba subsumes yours, or what the relation is in terms of eigenvalues of its state-transition matrix to yours?
> > >
> > > Mamba complex does not subsume AUSSM. The **complex Mamba version and our model are very different in terms of the eigenvalues of the state transition matrix**. In complex Mamba, the eigenvalue spectrum does not necessarily have to lie on the unit circle and the input can only scale these eigenvalues over time (through $\Delta_t$ in the Mamba paper). In AUSSM, the *eigenvalues are constrained to lie exactly on the unit circle in the complex plane, but the eigenvalues can change depending on input* and not just scale as is the case for Mamba-complex and the linked papers.
> > >
> > > ## ‘Further increase in formal expressivity is only possible using non-diagonal and, hence, a computationally inefficient approach.’
> > >
> > > Thank you for your comment, we agree that the nuances of this point should be clarified. In order to recognize all regular languages, under standard assumptions, *it is a necessary but not sufficient criterion to use non-diagonal transition matrices*.  This is because general transitions between n states beyond rotations and aperiodic ones require rank n matrix updates rather than the diagonal + rank-1 householder reflections used by DeltaNet. Further gains is expressivity may be possible with further generalization e.g. DeltaProduct [1].
> > >
> > > ## See Figure 3 in...
> > >
> > > The computational efficiency discussed in our paper is _asymptotically optimal_ from an algorithmic theory perspective due to the use of the work-efficient parallel scan algorithm (see the discussion in Appendix E.2) - meaning the number of operations required by the algorithm is only a constant factor of the sequential algorithm performing the computation. This is distinct from implementation-level optimizations, such as using hardware-specific CUDA Tensor Cores. While both approaches aim to improve runtime performance, their nature and generality differ. We elaborate on this distinction below and will include a more detailed discussion in the revised manuscript.
> > >
> > > **Optimization using Tensor Cores** - CUDA kernels can take advantage of tensor cores to achieve significant speedups, even for recurrent computations. High-performance implementations of xLSTM and Gated DeltaNet demonstrate that hardware-aware kernel optimizations using tensor cores can match or even exceed the runtime of asymptotically efficient parallel algorithms. However, there are limitations to this - Tensor Cores are currently efficient for half-precision (e.g., FP16) computations, and do not fully support full-precision (FP32) workloads. Their benefits are workload-dependent and do not offer a general substitute for algorithmic improvements that apply across hardware platforms and precision regimes.
> > >
> > > **Asymptotically Optimal** - The efficiency we demonstrate in our paper is _asymptotically optimal_ in the algorithmic sense: it is based on bounds for total work and parallel depth under standard models of computation and independent of specific hardware features. Unlike tensor cores that may yield high performance on currently available hardware, asymptotically optimal algorithms minimize total computation and communication overhead, making them applicable in more general cases.
> > >
> > > ## Would it be possible to provide throughput numbers compared to (Gated) DeltaNet?
> > >
> > > Unfortunately, it is not straightforward to obtain a fair comparison to Gated DeltaNet that uses tensor cores. The linked papers use hardware tricks that leverage half-precision FP16 operations, and fast recurrent tensor cores to obtain better performance compared to the mamba kernel. So, we cannot obtain an apples-to-apples comparison. It is an interesting future avenue to investigate whether the half precision FP16 tensor core version of AUSSM can match the performance of existing tensor core kernels.
> > >
> > > [1] Siems et al., 2025. DeltaProduct: Improving State-Tracking in Linear RNNs via Householder Products

---

> > > > ### Comment · Reviewer_MtYg · 2025-08-05
> > > >
> > > > I thank the authors for the clarifications. Thank you for providing background on the complex parameterization of Mamba and relating it to your own work.
> > > >
> > > > Regarding computational efficiency, I think this needs to be further clarified in the paper. You mentioned in a previous response 'computational efficiency' which one should interpret as the mix of algorithmic and hardware efficiency. Under this interpretation your statement is incorrect, because on current hardware it's much faster to opt for a chunk-wise form (as done by Mamba 2, DeltaNet etc) in order to leverage tensor-cores, rather than going for a selective-scan.
> > > >
> > > > I am raising my score to a weak accept.

---

### Official Review · Reviewer_MTfs · 2025-07-02

**Clarity:** 2
**Significance:** 2
**Originality:** 3
**Rating:** 4
**Confidence:** 3

**Summary:**

This paper proposes the Adaptive Unitary State Space Model (AUSSM), which is built upon a skew-symmetric ordinary differential equation. While AUSSM introduces a slight increase in computational complexity, the overhead is not significant. Compared to state space models (SSMs) with real-valued eigenspectra, AUSSM offers higher expressivity with unitary eigenspectra. In addition, the paper presents a hybrid architecture, AUSSM Hybrid, which integrates the AUSSM with Mamba blocks. The method primarily targets improvements in tasks where counting is crucial, and validation is conducted across various sequence modeling benchmarks.

**Questions:**

1. What is the theoretical or practical benefit of an SSM having the conservation property?
2. Aren’t complex-valued eigenvalues used not only in LinOSS but also in S4D and S5? In Mamba, complex-valued variants were also tested in ablation studies, and real-valued variants were selected due to slightly better performance. Emphasizing real eigenspectra as a prevailing trend in prior SSMs seems somewhat misaligned with the actual research landscape.
3. In the paper, Mamba is referred to as mildly adaptive, since it fixes $A$ (actually, it eventually varies through the use of $\Delta$), and adapts only $B$ and $C$. In some parts of the paper, however, Mamba is described as non-adaptive. Would it be more consistent to refer to it as *mildly adaptive*, and could a more detailed discussion on the role of $\Delta$ be included?
4. Can the baseline models (e.g., S5, S6, LinOSS, xLSTM, Mamba) and the proposed models (AUSSM, AUSSM Hybrid) be unified across all experiments? Is there a particular reason why different subsets of models are selected for different tasks?
5. The results on the UEA dataset differ significantly from those reported in the LinOSS paper. Can this be explained?
6. AUSSM Hybrid achieves the best performance, not the pure AUSSM model (especially in Tables 2 and 3). This raise concerns about whether AUSSM alone sufficiently supports the paper’s main claim of resolving the expressivity-efficiency tradeoff. If AUSSM indeed offers superior expressivity, shouldn't it also achieve strong results beyond counting-focused tasks?

**Ethical Concerns:**

["NO or VERY MINOR ethics concerns only"]

**Final Justification:**

I have raised my score, taking into account both the improvements in clarity and the remaining limitations due to the codebase.

**Limitations:**

yes

**Paper Formatting Concerns:**

There are no concerns regarding the paper's formatting.

**Quality:**

2

**Strengths And Weaknesses:**

- Strength
    - The paper provides a rigorous analysis of the underlying systems modeled by SSMs and explores a principled direction for improvement.
- Weaknesses
    1. The selection of prior works and baseline models in the experiments appears biased, especially in terms of real vs. complex eigenspectra.
    2. Since some experiments only report results for the hybrid model, it is difficult to isolate and evaluate the impact of the proposed AUSSM.

---

> ### Author Rebuttal · Authors · 2025-07-31
>
> We thank the reviewer for the critical evaluation of our work and for providing constructive criticism that ultimately improved the manuscript. Thank you for highlighting the paper **provides a rigorous analysis of the underlying systems modeled by SSMs and explores a principled direction for improvement** and for the comment that the overhead posed by introducing adaptive time varying matrices was not significant.
>
> ## Relevant Changes
>
> ### 1. Experiments with general eigenvalue baselines
>
> We added new results in the formal language benchmarks to compare with two versions of Mamba with general eigenvalue spectrum (NOTE: Mamba has eigenvalue range between \[0,1\])
>
> |                   | Mamba with Complex Eigenvalues | Mamba in \[-1,1\] range |
> | ----------------- | ------------------------------ | ----------------------- |
> | repetition             | 0.09                           | 0.1                     |
> | bucket sort          | 0.21                           | 0.91                    |
> | majority count         | 0.19                           | 0.31                    |
> | majority          | 0.13                           | 0.64                    |
> | solve equation             | 0.43                           | 0.24                    |
> | modarithmetic          | 0.12                           | 0.116                   |
> | modarithmetic wo bracket | 0.23                           | 0.24                    |
> | cyclenav          | 0.42                           | 0.91                    |
> | parity            | 0.27                           | 1.0                     |
>
> As reported in prior works \[1\], Mamba \[-1,1\] performs better compared to Mamba on the parity, cyclenav, solve equation, and bucketsort. However, AUSSM and the AUSSM Hybrid model still perform better in most tasks except bucketsort and solve equation. The tested complex version of Mamba performed poorly compared to the Grazzi version, which iterates the finding in the Mamba paper that real-eigenvalue parameterizations performed better. In AUSSM, the additionally introduced adaptivity and unitarity provide sufficient requirements for enabling learnability and expressivity.
>
> For the weather benchmark, we added S5 to our list of baselines
>
> | model | MAE  |
> | ----- | ---- |
> | S5    | 0.36 |
>
> S5 is found to have better MAE than Mamba and LinOSS, but our proposed AUSSM Hybrid model performs the best in the benchmark.
>
> ## Addressing Weaknesses and Questions
>
> ### Weakness 1: The selection of prior works and baseline models in the experiments appears biased, especially in terms of real vs. complex eigenspectra.
>
> To address this concern, we have added new results after running models on our suite of benchmarks. Now, our results for formal language benchmarks include Mamba (real), Mamba Complex (complex), and Mamba \[-1,1\] (\[1\]). For UEA benchmarks, we have Mamba (real), S5, S6, Linoss (Complex), and for the weather benchmark, we compare Mamba (real), S4, S5, LinOSS (complex)
>
> ### Weakness 2: Since some experiments only report results for the hybrid model, it is difficult to isolate and evaluate the impact of the proposed AUSSM.
>
> AUSSM Hybrid was chosen for real-world benchmarks since our theoretical analyses (Lemma 2) and empirical formal language results (Table 1) point to the AUSSM+S6 (mamba) hybrid model consistently performing better than a pure AUSSM block model. Note also that without the AUSSM block, the Hybrid model is a pure Mamba model for which results are already included in the tables.
>
> ### Question 1: What is the theoretical or practical benefit of an SSM having the conservation property?
>
> Conserved dynamics enable running SSMs on longer sequences without blowing up or decaying hidden states, and further enable stable gradient propagation during training. Prior to the introduction of SSMs in the recurrent neural network space, conserved dynamics were preferred in RNNs to avoid the diminishing/exploding gradient problems when processing long sequences (e.g. Arjovsky et al. (2016)). With the introduction of SSM initialization schemes such as HiPPO (Gu et al, 2020), such conservation schemes became less relevant since long-term stability was maintained through other means. However, newer SSMs like the LinOSS discussed in the paper still have conserved dynamics, suggesting that they help improve even newer SSMs to capture long-term dependencies.
>
> ### Question 2: Aren’t complex-valued eigenvalues used not only in LinOSS but also in S4D and S5? In Mamba, complex-valued variants were also tested in ablation studies, and real-valued variants were selected due to slightly better performance. Emphasizing real eigenspectra as a prevailing trend in prior SSMs seems somewhat misaligned with the actual research landscape.
>
> We have included new comparisons to S5, in UEA and weather, and Mamba with S4D initialization in formal language benchmarks (see Changes #1 above). Complex eigenvalue spectra have been used in many frontier SSM models - we discussed two main ones (LRU and LinOSS) in our background. However, the practice of real-valued eigenvalues is still a prevalent trend in SSMs, as evidenced by landmark papers - Mamba \[2\] rightly pointed out that real eigenvalues were better than complex ones, and MEGA \[3\] also shows similar results. The Mamba repository, which is widely adopted currently, also uses only positive real-valued recurrence (see open github issue #106 in the Mamba repo that shows that complex eigenvalue training is not supported out-of-the-box by the open source implementation).
>
> ### Question 3: In the paper, Mamba is referred to as mildly adaptive, since it fixes $A$ (actually, it eventually varies through the use of $\Delta$), and adapts only $B$ and $C$. In some parts of the paper, however, Mamba is described as non-adaptive. Would it be more consistent to refer to it as _mildly adaptive_, and could a more detailed discussion on the role of $\Delta$ be included?
>
> We apologize for the oversight in line 41 where we described Mamba as non-adaptive. **Mamba is partially (or mildly) adaptive** as rightly pointed out. We have corrected this. Mamba is mildly adaptive - i.e, the $A$ in the ODE formulation (eqn1) is time invariant, but the $A'$ in the discrete formulation (eqn 2) can vary based on the discretization timestep ($\Delta$). The role of $\Delta$ is only to scale the dynamics (fast or slow) depending on input, but if the nature of dynamical behavior needs to be changed - e.g., rotate left instead of right, or convergent instead of divergent, then the matrix $A$ itself needs to change. AUSSM is thus fully adaptive in the sense that you can control the dynamical behavior to that level of granularity.
>
> ### Question 4: Can the baseline models (e.g., S5, S6, LinOSS, xLSTM, Mamba) and the proposed models (AUSSM, AUSSM Hybrid) be unified across all experiments? Is there a particular reason why different subsets of models are selected for different tasks?
>
> The baselines were chosen depending on the currently reported state-of-the-art results in the respective tasks that were tested. To balance the choice of baselines, we have added additional results for S5, Mamba\[-1,1\] MambaComplex in the formal language benchmarks and S5 to the weather benchmark (see Changes #1 above). There are technical limitations that prevent rerunning certain models like LinOSS that are written in JAX because our code base is Pytorch. There is a nontrivial effort required to do the porting, and the significant risk of introducing silent bugs that may skew the results. With the newly introduced results, we have covered SSMs that use different parts of the eigenvalue spectrum.
>
> ### Question 5: The results on the UEA dataset differ significantly from those reported in the LinOSS paper. Can this be explained?
>
> **The results are the same as reported in the LinOSS paper**, the values we report are the scaled accuracies instead of the raw accuracies. Scaled accuracy scales the raw accuracy between baseline performance (scaled accuracy=0) and perfect performance (scaled accuracy=1). It is computed as (raw_accuracy - baseline)/(1-baseline) where baseline is the performance of a random estimator = (1/num_classes)
>
> ### Question 6: AUSSM Hybrid achieves the best performance, not the pure AUSSM model (especially in Tables 2 and 3). This raises concerns about whether AUSSM alone sufficiently supports the paper’s main claim of resolving the expressivity-efficiency tradeoff. If AUSSM indeed offers superior expressivity, shouldn't it also achieve strong results beyond counting-focused tasks?
>
> The claim is not that AUSSM has superior expressivity by itself, but rather that it adds a different kind of expressivity to the hybrid architecture. We have adjusted the diagram in Fig 1. to make this clear. AUSSM, however, takes a step toward general LTV systems which were not possible without sacrificing computational efficiency. In formal representation theory perspective, the complex unitary eigenvalues of AUSSM give it the ability to perform cyclic group operations, whereas Mamba components can perform the operations of aperiodic transformation monoids. Together, this gives the hybrid architecture the expressivity required to model solvable monoids, or equivalently, solvable languages. The Hybrid model achieves maximal expressivity in the class of computationally efficient diagonal SSMs, any more increase in expressivity require relaxing the diagonal constraint.
>
> ## References
>
> \[1\] Grazzi et. al. Unlocking State-Tracking in Linear RNNs Through Negative Eigenvalues. ICLR (2025)
> \[2\] Gu and Dao. Mamba 2023: Linear-time sequence modeling with selective state spaces. COLM (2024)
> \[3\] Ma et. al. Mega: Moving Average Equipped Gated Attention. ICLR (2023)

---

> > ### Comment · Reviewer_MTfs · 2025-08-06
> >
> > I appreciate the authors' efforts to clarify my concerns through additional experiments and explanations regarding how the methods demonstrate different advantages in hybrid models. The authors have addressed my questions well, and the additional experiments help improve the completeness of the empirical validation.
> >
> > **Follow-up to Q6:** Regarding the adjustment in Fig 1, does this imply that the authors now represent the expressivity of AUSSM and S6 to be at a similar level, and that the hybrid model is positioned higher to reflect its enhanced expressivity?

---

> > > ### Author Response · Authors · 2025-08-07
> > >
> > > We thank the reviewer for the positive comments on our new results and clarifications of the advantages of each AUSSM, and the hybrid AUSSM.  To address the Q6 follow-up, we add a revised Figure 1.A where the AUSSM and S6 are shown to solve different parts of a common class of solvable monoids, and AUSSM hybrid that combines S6 and AUSSM spans the entire class of solvable monoids, clearly increasing the expressivity of either model alone.

---

> > > > ### Comment · Reviewer_MTfs · 2025-08-07
> > > >
> > > > The authors' response helped clarify the revised figure. Although additional experiments have been provided in the rebuttal, some limitations remain unresolved.
> > > >
> > > > Taking this into account, I am raising my score to Weak Accept.

---

### Official Review · Reviewer_A2pm · 2025-07-16

**Clarity:** 3
**Significance:** 3
**Originality:** 4
**Rating:** 5
**Confidence:** 4

**Summary:**

The paper introduces Adaptive Unitary State Space Model (AUSSM), a sub class of general SSMs to find a middle ground between theoretical expressivity limitations of the architecture and the performance / scalability on downstream tasks. The recurrent matrix $A_t$ is similar to Mamba styled models (S6), where the matrix is input dependent. Unlike Mamba, the authors here propose to make the matrix skew symmetric, as well for the overall discrete product $exp(\Delta A)$to have Eigen values lie on a unit circle. This change ensures stability (avoiding exploding / vanishing gradients), while also allowing for separable kernel formulation. The authors also provide a CUDA kernel implementation to make these computations faster. The authors test this AUSSM block in isolation, or in a hybrid setup along with a Mamba block as well. Theoretical analysis of these reveals that the hybrid setup overcomes some of the theoretical limitations of the original Mamba block, specially for modular counting (for eg. Parity, which Mamba provably can't solve at arbitrary lengths, but AUSSM blocks can). To complement the theory, the authors conduct some experiments on the formal languages for which their claims were made as well as on some time series dataset to show that their architecture can be scaled to model long term dependencies as well.

**Questions:**

1. **Hybrid performance dips**
Table 1 shows a few tasks where the AUSSM + Mamba hybrid lags behind either component on its own, even though Note 1 (lines 619–632) argues the behavior of hybrid models can be made identical to either a Mamba / AUSSM block on its own. Could you provide a bit more detail about what you think could be the reason behind slightly worse performance on these tasks vs the others.

2. **Neuroscience connection**
The connection to the Neuroscience literature to me seemed not very clear / very limited. Is the connection to the neuroscience literature restricted to the fact some jPCA procedure where such skew symmetric ODE is used or is the derivation of the AUSSM block also inspired from it ?

3. **Contextualizing results compared to recent SSM advancements** As mentioned in weaknesses, other recent work [1], [2] has also aimed to enhance SSM expressivity. How does AUSSM compare to these, could you contextualize your contributions vis a vis theirs ?

*(Side note: I was able to check most appendix material, but Section F is the only section I couldn't verify properly, because of lack of background knowledge on my part there)*

[1] Grazzi et al 2025, Unlocking State-Tracking in Linear RNNs Through Negative Eigenvalues, ICLR 2025.
https://openreview.net/pdf?id=UvTo3tVBk2

[2] Peng et al, 2025 RWKV-7 "Goose" with Expressive Dynamic State Evolution, Arxiv
https://arxiv.org/pdf/2503.14456

**Ethical Concerns:**

["NO or VERY MINOR ethics concerns only"]

**Final Justification:**

I am satisfied with the authors responses, and if they include the details about the new experiments (that also take into account, other similar changes proposed by other recent work in this area) in the main paper, I believe this would be a good contribution to the SSM architecture landscape.

**Limitations:**

yes

**Quality:**

4

**Strengths And Weaknesses:**

Strengths
- The paper contributes to the space of SSMs by incorporating small changes in the parameterization of an SSM and thus achieving a theoretically more expressive architecture. All the Theorems and Lemmas seem correct. I was able to verify most proofs. (see questions for things I didn't understand).
- The paper contributes both on the theoretical side, as well as provides efficient CUDA Kernel implementations to practical relevance.
- Adequate empirical evidence for a theory paper. Algorithmic tasks confirm the formal-language claims, and the UEA/Weather benchmarks show the gains transfer to realistic long-range settings, all on modest hardware.

Weaknesses
- The only relevant weakness I could think of was -- inadequate comparison / discussion about similar changes proposed by some of the other recent works. For example the papers - [1], [2], achieve somewhat similar results, compared to what the authors get. I can intuitively feel that the changes the authors are proposing here, should be even better than the change proposed at least in [1], but I am not sure about [2], so it would be good from the authors point of view, to contextualize their contribution compared to these other ones.
- Minor presentation issues (just a subset, requesting the authors to do a better passthrough for the next version) .
Typos: Line 63 “Crdinary → Ordinary”; Line 67 “respectivelt → respectively”; Line 69 and Line 196 difficult to parse.
- Missing citations: The appendix references papers 36, 37, … that are absent from the main list (only 1–33 appear).
This made understanding some of the points unclear. For example, I am unfamiliar with some of the precision results cited, Line 598 - 599 -I am unfamiliar with the paper that states that complex multiplication incurs an error $\lt \sqrt{5} \epsilon$, and so couldn't verify what the authors meant there. Maybe put those additional references in the rebuttal response, so that it's easier to understand the results authors were referring to.

[1] Grazzi et al 2025, Unlocking State-Tracking in Linear RNNs Through Negative Eigenvalues, ICLR 2025.
https://openreview.net/pdf?id=UvTo3tVBk2

[2] Peng et al, 2025 RWKV-7 "Goose" with Expressive Dynamic State Evolution, Arxiv
https://arxiv.org/pdf/2503.14456

---

> ### Author Rebuttal · Authors · 2025-07-31
>
> We thank the reviewer for comments and suggestions for improving the manuscript and highlighting our **core contributions to the theory and a computationally efficient CUDA kernel implementation**. We thank the reviewer for also highlighting that the benchmarks confirm our theoretical predictions, and the real-world settings also adequately show performance improvements.
>
> ## Relevant Changes
>
> ### 1. Experiments with general-eigenvalue baselines
>
> We thank the reviewer for providing relevant works. We added new results in the formal language benchmarks to compare with two versions of Mamba with general eigenvalue spectrum (NOTE: Original Mamba has eigenvalue range between \[0,1\])
>
> |                   | Mamba with Complex Eigenvalues | Mamba in \[-1,1\] range |
> | ----------------- | ------------------------------ | ----------------------- |
> | repetition             | 0.09                           | 0.1                     |
> | bucket sort          | 0.21                           | 0.91                    |
> | majority count         | 0.19                           | 0.31                    |
> | majority          | 0.13                           | 0.64                    |
> | solve equation             | 0.43                           | 0.24                    |
> | modarithmetic          | 0.12                           | 0.116                   |
> | modarithmetic wo bracket | 0.23                           | 0.24                    |
> | cyclenav          | 0.42                           | 0.91                    |
> | parity            | 0.27                           | 1.0                     |
>
> As rightly pointed out, Mamba \[-1,1\] performs better compared to Mamba on the parity, cyclenav, solve equation, and bucketsort. However, AUSSM and the AUSSM Hybrid model still perform better in most tasks except bucketsort and solve equation. The tested complex version of Mamba performed poorly compared to the Grazzi version, which iterates the finding in the Mamba paper that real-eigenvalued parameterizations performed better. In AUSSM, the additionally introduced adaptivity and unitarity provide sufficient requirements for enabling learnability and expressivity.
>
> For the weather benchmark, we added S5 to our list of baselines
>
> | model | MAE  |
> | ----- | ---- |
> | S5    | 0.36 |
>
> S5 is found to have better MAE than Mamba and LinOSS, but it our proposed AUSSM Hybrid model performs the best in the benchmark.
>
> ## Addressing Weaknesses and Questions
>
> ### Weakness 1: The only relevant weakness I could think of was -- inadequate comparison / discussion about similar changes proposed by some of the other recent works. For example the papers - \[1\],\[2\], achieve somewhat similar results, compared to what the authors get. I can intuitively feel that the changes the authors are proposing here, should be even better than the change proposed at least in\ [1\], but I am not sure about \[2\], so it would be good from the authors point of view, to contextualize their contribution compared to these other ones.
>
> Thank you for the suggestion. We have added the modified Mamba implementations (Mamba\[-1, 1\] Grazzi, Mamba Complex) in the formal language evaluations. Note that negative eigenvalues only give a small expressivity benefit over non-negative eigenvalues. The results we obtained (see Changes #1 above) are consistent with the prior results - Mamba\[-1,1\] improves on the base Mamba for most tasks but falls behind AUSSM hybrid in some. Mamba Complex, however, performs poorly in most tasks compared to both Mamba\[-1,1\] and AUSSM hybrid despite higher expressivity. This contradiction was found in the original Mamba paper, too. It is possible that Mamba with pure complex spectrum may have poorer learnability for many tasks - there are no free lunches.
> RWKV-7 is an LRNN that does not have a diagonal transition matrix, which in theory could make it more expressive than our architecture in certain settings, though at the price of additional computation cost.
>
> ### Weakness 3: Minor presentation issues (just a subset, requesting the authors to do a better passthrough for the next version). Typos: Line 63 “Crdinary → Ordinary”; Line 67 “respectivelt → respectively”; Line 69 and Line 196 difficult to parse.
>
> We thank the reviewer for pointing these out. These are now corrected in the manuscript.
>
> ### Weakness 4: Missing citations: The appendix references papers 36, 37, … that are absent from the main list (only 1–33 appear). This made understanding some of the points unclear. For example, I am unfamiliar with some of the precision results cited, Line 598 - 599 -I am unfamiliar with the paper that states that complex multiplication incurs an error, and so couldn't verify what the authors meant there. Maybe put those additional references in the rebuttal response, so that it's easier to understand the results authors were referring to.
>
> We apologize for this oversight. The missing citations are:
> \[34\] Babcsányi. Automata with finite congruence lattices. Acta Cybernetica, 2007.
> \[35\] Salkoff et al. Propagation of traveling waves in the visual cortex requires microcircuit coordination. Neuron, 2020.
> \[36\] Brent et al. Error bounds on complex floating-point multiplication. Mathematics of Computation, 2007.
> \[37\] Zimmermann. On krohn-rhodes theory for semiautomata.
> \[38\] Hewitt et al. RNNs can generate bounded hierarchical languages with optimal memory. EMNLP, 2020
> \[39\] Cormen et al. Introduction to Algorithms, Third Edition. The MIT Press, 2009.
>
> ### Question 1: Hybrid performance dips
>
> We fix the number of blocks as two and try 4 different combinations - AM MA AA MM. A: AUSSM block and M: Mamba Block. The pure AUSSM (AA) is a good middle ground, but not the most expressive possible - see our theoretical analysis (Lemma 1 and Lemma 2 in the paper). As such, there may be tasks that require two layers of mamba for non-unitary recurrence or full adaptivity. The details of why exactly this contradiction is present are not clear at the moment and are described as one of the limitations of the current study.
>
> ### Question 2: Neuroscience connection
>
> The AUSSM ODE is derived from the jPCA procedure (detailed in appendix). The jPCA procedure, introduced by Churchland et al. (2012), was designed to study rotational dynamics in neural activity, and has been used widely in neuroscience to uncover such dynamics from neural recordings during tasks such as motor planning and execution. By fitting low-dimensional projections that maximize antisymmetric structure in the population dynamics, jPCA reveals patterns consistent with conservative, unitary evolution. Finding such conservative dynamics strongly represented in real neural data suggests that portions of cortical activity evolve with minimal dissipation or loss of energy -- exactly analogous to the AUSSM, and thereby inspiring both the method and the goal. The block structure is not biologically inspired but is used in the Mamba class of models.
>
> ### Question 3: Contextualizing results compared to recent SSM advancements
>
> \[1\] and \[2\] both use non-diagonal recurrences, sacrificing some computational efficiency for enhanced expressivity, allowing them, in theory, to recognize all regular languages. Note, however, that a single Householder reflection per step is insufficient for full regular language expressivity or even to perform arbitrary rotations. As \[1\] points out, one needs products of multiple reflections to realize arbitrary orthogonal transformations, which can be costly. By contrast, AUSSM can perform rotations cheaply with a single diagonal update. Our contribution, therefore, is making *diagonal SSMs* maximally formally expressive while keeping computational overhead small, extending their expressivity to all solvable languages (a subset of the regular languages), which is also widely believed to be the upper expressivity bound for transformers \[3\]. Due to time limitations, we have added only results for running \[1\] on the formal language benchmarks, though we do compare to xLSTM, which is also non-diagonal like \[2\], but is additionally a non-linear RNN. We have added a related work section to the appendix contextualizing our expressivity results with regard to other approaches, which unfortunately does not fit within the character limit of our response.
>
> ## References
>
> \[1\] Grazzi et al., 2025. Unlocking State-Tracking in Linear RNNs Through Negative Eigenvalues. ICLR (2025)
> \[2\] Peng et al., 2025. RWKV-7 "Goose" with Expressive Dynamic State Evolution
> \[3\] Merrill et al., 2022. Saturated Transformers are Constant-Depth Threshold Circuits

---

> > ### Comment · Reviewer_A2pm · 2025-08-01
> >
> > I thank the authors for their detailed response.
> > The new results that engage with previous literature that I pointed out would be good. I request the authors to include that in the new version of the manuscript. Additionally a more detailed discussion engaging with the recent SSM advancements would also be good to include in the paper. The neuroscience connection explained by the authors in this response helped me understand the motivation a bit better.
> >
> > Overall I am happy with the response, and would like to keep my score.

---

### Note · Authors · 2025-08-13

We thank all the reviewers for constructive feedback for the improvement of the manuscript and for recognizing that AUSSM **generalizes the eigenvalue spectrum** of partially-adaptive diagonal SSMs (used in Mamba) to fully adaptive diagonal SSMs in AUSSM. We further thank the reviewers for highlighting our contributions to the **theory of SSMs with rigorous proofs of expressivity, the practical relevance through an efficient CUDA kernel implementation and empirical evidence to support the claims**. We are happy to note that we have addressed the questions relating to experimental support, clarity and related works during the review period and 2 reviewers MtYg, MTfs **raised their scores in response to these changes**. We summarize the changes below:

1. Added additional empirical support on the performance of partially adaptive S6 with negative and complex eigenvalue spectra. This extends the comparison we have of AUSSM to general eigenvalue distributed SSMs.
2. Added a new section in the appendix to discuss relevant related works extending the spectra, and non-linear unitary RNNs.
3. Clarifications regarding the performance results of AUSSM Hybrid model, the connection to neuroscience.
4. Clarification of the optimization claims and a note about comparison with alternate Tensor core optimization approaches. We highlight that our proposed algorithm still tackles the most general case.
5. Clarification of expressivity in our discussion regarding generalization to non-diagonal matrices.

---

### Decision · Program_Chairs · 2025-09-17

**Decision:**

Accept (poster)

**Comment:**

The paper proposes a modification to SSM they call AUSSM which uses linear time varying recurrence and a unitary eigenvalue spectrum derived from a skew-symmetric ODE. The authors show that this modification allows hybrid models Mamba + AUSSM to represent all solvable automata, here the negative eigenvalue helps implement the counter automata. The authors additionally provide a separable convolution formulation and a CUDA implementation to efficiently implement AUSSMs. Experiments support the benefits of the hybrid models compared to other variants across simple arithmetic tasks and long time-series modeling.

The paper was generally well-received by the reviewers, and the authors cleared most of the concerns of the reviewers with additional experimentation and exposition over the rebuttal. While the impact of the proposed modification is unclear for more general tasks and at larger scale, and whether the computational cost would be justified for expressivity benefits, this paper still adds a novel result in the area of improving SSMs that would be of interest to the NeurIPS audience. Therefore I lean towards accepting the paper. I encourage the authors to include the additional experiments and discussion that happened during the rebuttal period, and improve the exposition of the paper.